# Auxin-inducible degradation of UNC-116 in *C. elegans* inhibits bidirectional dense core vesicle transport and worm locomotion on different timescales

Astrid H. Boström[1,‡], Anna Gavrilova[1,2,*], James C. Herzig[3], Gino B. Poulin[1] and Victoria J. Allan[1,‡]

## ABSTRACT

The microtubule motor kinesin-1 is vital in neurons, with mutations being associated with neurological diseases. Deletion of the *Caenorhabditis elegans* kinesin-1 gene *unc-116* is lethal, and viable mutants are uncoordinated. Here, we perform auxin-mediated degradation of the UNC-116 protein and monitor effects on worm locomotion and dense core vesicle (DCV) transport. UNC-116 is degraded within 1–3 h of auxin treatment. Impaired swimming and crawling are detected after 6–14 h, depending on organismal age, and degradation of UNC-116 in neurons alone disrupts locomotion. Effects on DCV transport are observed sooner, with 4 h of degradation strongly inhibiting both anterograde and retrograde movement. This motility recovers when worms are transferred to auxin-free plates for 24 h, even though protein level recovery is incomplete. Rescue from a 10 h auxin treatment completely restores crawling speed, but although swimming is initially unaffected, it becomes strongly inhibited during the rescue period, suggesting that UNC-116 has distinct roles in these locomotive gaits. Overall, by bypassing early developmental UNC-116 functions, we reveal that UNC-116 is essential for bidirectional DCV transport and crucial for worm locomotion.

KEY WORDS: Kinesin, Dense core vesicle, Neuron, *C. elegans*, Degron, Locomotion

## INTRODUCTION

Intracellular transport of cargoes along microtubules is fundamental for maintaining cellular organisation (Cason and Holzbaur, 2022; Sleigh et al., 2019). This process is carried out by two classes of microtubule motors – kinesins and cytoplasmic dynein (dynein). Most kinesins are anterograde motors, transporting cargoes toward the microtubule plus-ends, whereas dynein transports cargoes retrogradely, toward minus-ends (Hirokawa et al., 2009;

[1]School of Biological Sciences, Faculty of Biology, Medicine and Health, University of Manchester, The Michael Smith Building, Rumford St, Manchester M13 9PT, UK. [2]Department of Mathematics, Faculty of Science and Engineering, The University of Manchester, Manchester M13 9PL, UK. [3]MRC-University of Glasgow Centre for Virus Research, Garscube Campus, Switchback Road, Glasgow G61 1AF, UK. *Present address: Cell and Developmental Biology, University College London, Gower Street, London, WC1E 6BT, UK.

[‡]Authors for correspondence (viki.allan@manchester.ac.uk; astrid.bostrom@manchester.ac.uk)

A.H.B., 0000-0003-0329-6305; A.G., 0000-0003-2278-7820; J.C.H., 0000-0001-6434-4150; G.B.P., 0000-0003-1016-348X; V.J.A., 0000-0003-4583-0836

Reck-Peterson et al., 2018). Intracellular transport is particularly crucial in neurons as components must be transported long distances to reach the length of axons and dendrites (Bentley and Banker, 2016; Cason and Holzbaur, 2022; Guedes-Dias and Holzbaur, 2019).

Kinesin-1, one of the best characterised motors, is responsible for carrying a diverse range of cargoes, such as membranous cargo, mRNA and microtubules themselves (Cross and Dodding, 2019; Miki et al., 2005; Verhey and Hammond, 2009). Mammals have three genes encoding the kinesin-1 motor subunit – the ubiquitous *KIF5B* and the neuron-specific *KIF5A* and *KIF5C* (Kanai et al., 2000). Kinesin-1 mutations are associated with disorders including amyotrophic lateral sclerosis, Charcot–Marie–Tooth disease and hereditary spastic paraplegia, leading to eventual spasticity and/or movement difficulties (e.g. Baron et al., 2022; Bayrakli et al., 2015; Chiba and Niwa, 2024; Melo et al., 2015; Nam et al., 2018).

The nematode *C. elegans* has one isoform of the kinesin-1 motor subunit, UNC-116. Complete loss of UNC-116 function by gene knockout is embryonically lethal, and hypomorphic *unc-116* mutants, where the UNC-116 protein retains some function, are uncoordinated and move very poorly (Gavrilova et al., 2024; Patel et al., 1993; Sakamoto et al., 2005; Yang et al., 2005). This reduced locomotion is not likely to involve defects in synaptic vesicle delivery, as they are transported by the kinesin-3 member UNC-104 (Hall and Hedgecock, 1991; Kumar et al., 2010; Niwa et al., 2016; Ou et al., 2010; Wu et al., 2013; Zheng et al., 2014). However, kinesin-1 cargoes with a potential role in maintaining locomotion include mitochondria (Chen et al., 2021; Rawson et al., 2014; Sure et al., 2018; Zhao et al., 2021), glutamate receptors (Hoerndli et al., 2013, 2015) and dense core vesicles (DCVs) (Gavrilova et al., 2024). *unc-116* mutant animals also have developmental problems, including microtubule polarity reversals in dendrites (Harterink et al., 2018; He et al., 2020; Yan et al., 2013) and defects in axonal outgrowth and branching (Aguirre-Chen et al., 2011; Drozd and Quinn, 2023; Gavrilova et al., 2024; Su et al., 2006). The severe locomotion defects seen in *unc-116* mutants could therefore be due to the reduced movement of multiple cargoes combined with problems in neuronal development.

Neuronal DCVs deliver and release neuropeptides, neurotrophic factors and other signalling components (Gondré-Lewis et al., 2012). This delivery contributes to many aspects of physiology, including learning, development, locomotion and ageing (Gondré-Lewis et al., 2012; Randi et al., 2023; Ripoll-Sánchez et al., 2023). DCV motility is complex, as they are transported bidirectionally in axons by a combination of kinesin-1, kinesin-3 and dynein (Barkus et al., 2008; Gavrilova et al., 2024; Gumy et al., 2017; Kwinter et al., 2009; Lim et al., 2017). Evidence suggests that transport out of the cell body is carried out by the faster kinesin-3, after which kinesin-1 drives continued transport to the axon tip (Gumy et al., 2017; Lim et al., 2017; Park et al., 2023). Compromising one motor often leads to defective transport in both directions rather than upregulating

unidirectional movement, suggesting that antagonistic motors are somehow dependent on each other (Ally et al., 2009; Gavrilova et al., 2024; Hancock, 2014; Kwinter et al., 2009; Lim et al., 2017; Pilling et al., 2006; Yi et al., 2011). The mechanisms governing this 'paradox of co-dependence' remain unclear (Hancock, 2014). Investigating the roles of specific microtubule motors in multicellular models is needed to shed light on how bidirectional transport is regulated and enhance our understanding of how dysfunction leads to disease phenotypes.

Depleting the UNC-116 protein at specific times during development and once development is complete offers a way to investigate the multifaceted functions of kinesin-1. To do this, we have made use of the auxin-induced degradation (AID) system, where a protein tagged with a degron amino acid sequence is rapidly degraded upon auxin addition by a plant TIR1 F-box protein that forms an SCF E3 ubiquitin ligase complex with endogenous Skp1 and Cullin (Nishimura et al., 2009). In *C. elegans*, conditional and tissue-specific protein degradation can be mediated when worms expressing TIR1 in specific cells are exposed to auxin (Ashley et al., 2021; Zhang et al., 2015). AID has been used to degrade the dynein motor subunit (Cavin-Meza et al., 2022; Zhang et al., 2015), and the kinesin-3 motor UNC-104 (Cahoon and Libuda, 2021). Very recently, UNC-116 AID has been used to analyse the role of kinesin-1 in the positioning of mitochondria (Wu et al., 2024) and spectrin (Glomb et al., 2023) in the DA9 neuron, and kinesin light chain AID has validated the involvement of kinesin-1 in the outward movement of the spindle in meiosis I (Aquino et al., 2025).

Here, we characterise the effect of UNC-116 degradation on physiology and cargo transport by monitoring worm locomotion and analysing DCV motility. UNC-116 protein levels drop after 1–3 h of auxin treatment. There is a delay between protein loss and the onset of locomotion defects, which do not appear until 6–14 h of degradation. Importantly, degrading UNC-116 specifically in neurons is sufficient to give an uncoordinated phenotype. Moreover, UNC-116 loss has distinct effects on crawling and swimming over time, dependent on developmental stage, highlighting that the system can be used to provide insights about UNC-116 involvement in different neuromuscular processes. Effects on cargo transport appear sooner after protein loss, and bidirectional DCV movement in the ALA interneuron is affected after 1 h and substantially inhibited after 4 h. After 24 h of degradation, steady-state DCV distribution along the ALA axon is significantly altered. The effects of UNC-116 loss on DCVs appear to be global and cargo specific, as DCV distribution and transport is also affected in the axon of the DB7 motor neuron, but the distribution of synaptic vesicles remains unchanged. Thus, UNC-116 works as a crucial regulator of bidirectional DCV trafficking that, if lost, promptly halts transport. Finally, we test whether protein level, locomotion and DCV transport can recover following removal of worms from auxin and find that recovery of protein level and DCV transport is dependent on duration of original auxin treatment. Strikingly, however, worms can regain their ability to crawl but not swim, indicating that UNC-116 has distinct functions during these two gaits of motion. Overall, we demonstrate that conditional depletion of UNC-116 can be used to dissect its fundamental functions in cargo transport and physiology on a variety of timescales.

## RESULTS
### UNC-116 degradation is rapid and efficient
To establish whether acute AID-mediated degradation of UNC-116 was feasible, we used immunoblotting to monitor protein levels. An UNC-116 auxin-inducible degron strain was generated by inserting

the minimal AID* tag (Zhang et al., 2015) followed by an ALFA tag (Götzke et al., 2019) at the 5′ end of the endogenous *unc-116* gene. The AID*-tagged strain, referred to here as *unc-116(deg)*, was crossed with a strain expressing a single copy of the TIR1 F-box gene downstream of the ubiquitous promoter *eft-3* (Ashley et al., 2021) to give *unc-116(deg);uTIR1*. When grown in the presence of auxin, TIR1 associates with the AID* and leads to the ubiquitylation and proteasomal degradation of the target protein complex in all cells (Fig. 1A). We also crossed *unc-116(deg)* with a transgenic strain expressing TIR1 downstream of the pan-neuronal *rab-3* promoter *(nTIR1)* as an extrachromosomal array, given that neuronal degradation of UNC-116 might be a better model for phenotypes caused by the loss of function of the neuron-specific human isoforms of kinesin-1 heavy chain, *KIF5A* and *KIF5C* (de Ligt et al., 2012; Dutta et al., 2018; Ebbing et al., 2008; Liu et al., 2014; Poirier et al., 2013).

The UNC-116 protein was undetectable in *unc-116(deg);uTIR1* following treatment of L1 larvae with synthetic auxin (K-NAA) for 24 h (Fig. 1B) and greatly reduced when L1 worms developed to adults with K-NAA treatment for 72 h (Fig. 1C). A slight decrease in UNC-116 protein could be detected by immunoblotting following neuronal-specific degradation for 72 h in *unc-116(deg);nTIR1* worm, but this was not statistically significant (Fig. 1C). As AID systems can demonstrate basal degradation by TIR1 in the absence of auxin (Hills-Muckey et al., 2022; Kanke et al., 2011; Natsume et al., 2016; Negishi et al., 2022; Schiksnis et al., 2020), we compared UNC-116 levels in the absence of K-NAA in the *unc-116(deg)* strains with and without uTIR1 expression (Fig. 1B,C). Although the presence of uTIR1 reduced UNC-116 levels somewhat in young larvae without K-NAA, the difference was not statistically significant.

Next, we investigated the rate of ubiquitous UNC-116 degradation in L4 and adult worms. To do this, L1 worms were allowed to develop for 48 or 72 h, with K-NAA treatments starting 1, 3, 5 or 24 h before the end point (Fig. 1D). UNC-116 degradation was detected within 1 h, and appeared more rapid in L4 worms, suggesting that speed of degradation is influenced by life stage, as reported in Zhang et al. (2015). The small residual pool of UNC-116 that remained after 24 h of K-NAA treatment might correspond to the germline pool of UNC-116, which has important meiotic functions (Ellefson and McNally, 2009; McNally et al., 2012; Yang et al., 2005). The *eft-3* promoter driving TIR1 expression has been previously reported to express only at low levels in the germline, leading to a residual germline pool after degradation of the dynein heavy chain (Zhang et al., 2015).

Altogether, these results demonstrate that UNC-116 protein is efficiently and rapidly degraded in the presence of K-NAA in larval and adult worms, with statistically insignificant levels of background degradation without K-NAA.

### Swimming and crawling ability is drastically reduced by UNC-116 degradation
Worms with hypomorphic *unc-116* mutations are uncoordinated (Patel et al., 1993; Yang et al., 2005). The hypomorphic *unc-116(rh24sb79)* mutant has markedly reduced ability to crawl on stiff agar and swim in liquid (Gavrilova et al., 2024). These locomotion defects could be due to direct effects on cargo transport in neurons or muscle cells. However, the presence of the partially compromised motor in all cells from birth might affect microtubule organisation or have pleiotropic effects on neuronal development that affect their function. Thus, we reasoned that degrading UNC-116 in all cells should lead to significant crawling and swimming defects, providing a method for validating degradation efficiency phenotypically.

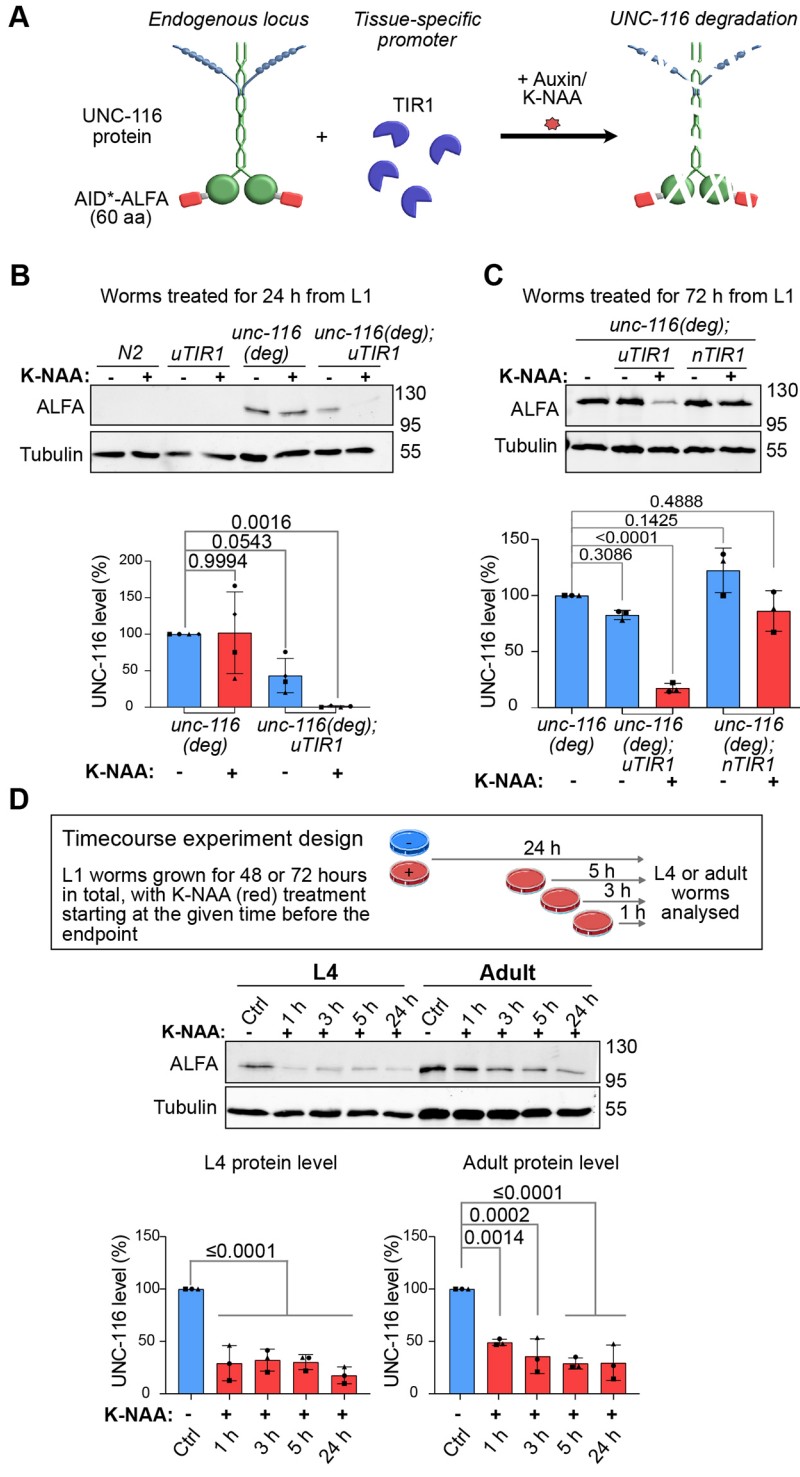

**Fig. 1. Robust UNC-116 degradation is seen upon K-NAA treatment.** (A) Schematic of UNC-116 degradation using the AID system. UNC-116 is shown in green, and kinesin light chains are in blue. (B,C) Synchronised L1 worms of the indicated strains were grown with or without K-NAA for 24 h (B) or 72 h (C). Immunoblotting was used to assess UNC-116 degradation using anti-ALFA antibodies with anti-tubulin as a loading control. Quantification was normalised to *unc-116 (deg)* (-) and anti-tubulin. (D) Time-course of UNC-116 protein degradation in L4 larvae and adult *unc-116(deg); uTIR1* worms grown with K-NAA for the indicated times compared with untreated worms. Means±s.d. are shown (*N*=4 independent experiments for B, *N*=3 for C and D) with *P*-values from one-way ANOVA followed by post-hoc Dunnett test.

We compared the swimming (also called 'thrashing') and crawling ability of adults that were allowed to develop from L1 for 72 h (to adulthood) on plates with or without K-NAA (Movies 1 and 2). The number of body bends per second (a measure of swimming ability) and average crawling speed of *unc-116(deg); uTIR1* was drastically reduced upon UNC-116 degradation, to the same extent as in the *unc-116(rh24sb79)* mutant (Fig. 2A,B). Importantly, there was no significant difference in the swimming and crawling ability of *unc-116(deg)* worms compared to that of wild-type worms (*N2*), indicating that the AID*-ALFA tag on UNC-116 protein did not affect locomotion. However, *unc-116(deg);uTIR1* worms raised on plates *without* K-NAA had reduced swimming ability compared to that of *N2*, but normal crawling speed (Fig. 2A,B). Thus, the effects of low levels of basal UNC-116 degradation by TIR1 in the absence of K-NAA could be detected by the reduced swimming but not crawling ability.

We next asked whether initiating UNC-116 degradation in L4 and adult worms affected locomotion, and if so, how long it took for the locomotion phenotypes to manifest. We performed time-course studies of crawling and swimming in *unc-116(deg);uTIR1* L4 and

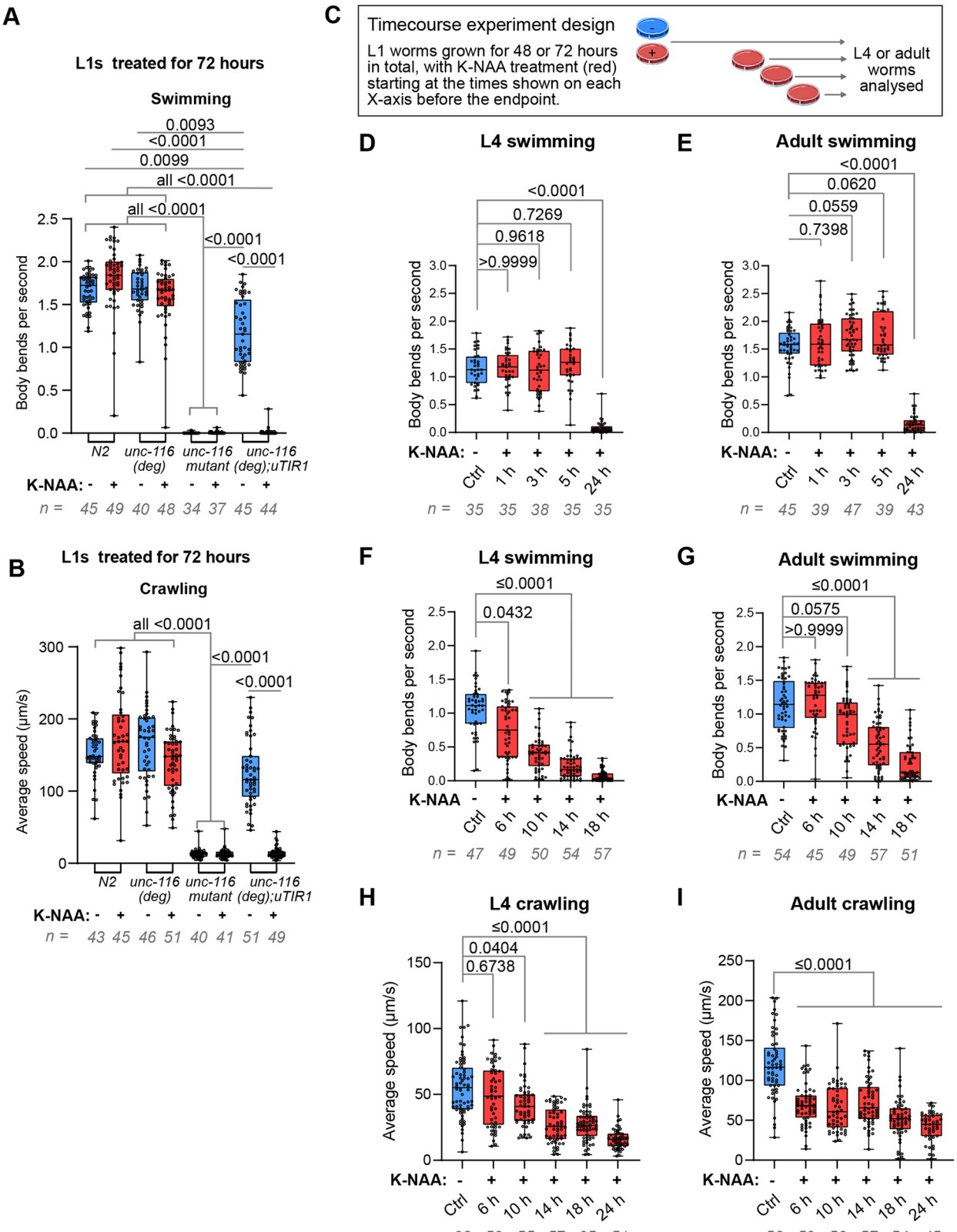

**Fig. 2. Effects of UNC-116 degradation on worm swimming and crawling.** (A) Swimming ability (body bends per second) and (B) crawling ability (average speed) of *N2*, *unc-116(deg)*, *unc-116(rh24sb79)*, and *unc-116(deg);uTIR1* adults grown with or without K-NAA for 72 h after L1 synchronisation. (C) Experimental design of time-course analysis of locomotion. Swimming ability measured as body bends per second, in L4s (D) or adults (E) using the same time points as for Fig. 1D. Expanded time-course analysis of swimming ability (F,G) and crawling speed (H,I), for L4 worms (F,H) and adults (G,I). All graphs show combined data from three biological repeats, with the total number of worms analysed indicated below each graph. Boxes display the median and interquartile range with whiskers from the minimum to the maximum value. Statistical analysis was by Kruskal–Wallis test followed by post-hoc Dunn's test. Only *P*-values ≤0.05 are indicated on the graphs in A and B.

Journal of Cell Science

adult worms treated with K-NAA for various times (Fig. 2C). No significant change in swimming ability was seen after 1–5 h of K-NAA treatment in adult and L4 worms (Fig. 2D,E), even though UNC-116 protein was degraded (Fig. 1D). After 24 h of treatment, however, swimming ability was strongly reduced (Fig. 2D,E). By assaying additional time points, we found that L4 worms showed a significant reduction in body bends per second after 6 h of K-NAA treatment, whereas for adults this was evident only after 14 h of treatment (Fig. 2F,G). When the effects on crawling were assessed, a somewhat different pattern was seen. A significant reduction in average crawling speed was already observed after 6 h of degradation in adults, but only after 10 h of degradation in L4 worms (Fig. 2H,I). These data show that being able to degrade UNC-116 over time at specific developmental stages can reveal unexpected differences in the sensitivity of behavioural traits to UNC-116 loss.

The above experiments used worms expressing TIR1 in all cells. To test whether degrading UNC-116 in neurons alone was sufficient to cause locomotion defects, we carried out swimming and crawling assays in adult *unc-116(deg);nTIR1* worms (Fig. 3A,B; Movies 3 and 4). Strikingly, treatment with K-NAA for 24 h (from the L4 stage to adulthood) resulted in a markedly reduced swimming and crawling ability, like that seen in *unc-116(deg);uTIR1* worms. Interestingly, the effect of background degradation on swimming seen with ubiquitous TIR1 expression was not observed with neuronal TIR1 (Fig. 3A). Altogether, these data indicate that UNC-116 function in neurons is necessary for normal locomotion.

### Characterisation of bidirectional DCV transport in *unc-116* degron strains without K-NAA treatment

Kinesin-1 has a wide range of cargoes that might contribute to the neuronal control of worm swimming and crawling, including mitochondria, glutamate receptors and dense core vesicles (Chen et al., 2021; Ding et al., 2022; Gavrilova et al., 2024; Hoerndli et al., 2013, 2015; Rawson et al., 2014; Sure et al., 2018; Zhao et al., 2021).

We chose DCVs as an exemplar cargo to assess the effects of UNC-116 degradation on cargo transport as we have previously demonstrated that DCV movement in both directions is profoundly inhibited in the ALA interneuron in the hypomorphic *unc-116(rh24sb79)* mutant (Gavrilova et al., 2024). To test the effect of acute UNC-116 loss on DCV movement using AID, we crossed the *unc-116(deg), unc-116(deg);uTIR1* and *unc-116(deg);nTIR1* strains with the *ida-1::gfp* strain, which expresses the GFP-tagged DCV transmembrane protein IDA-1 in the ALA, VC, HSN and PHC neurons (Cai et al., 2004; Zahn et al., 2001, 2004). The ALA is ideal for imaging as it has its cell body in the head and two laterally symmetrical axons running the length of each side of the body, with a further short neuronal process in the head (Fig. 4) (Ramirez-Suarez et al., 2019; Sanders et al., 2013).

Spinning disc confocal recordings of DCV movement were taken in both the proximal and distal regions of the axon. As previously, DCVs were observed as either motile or stationary (de Wit et al., 2006; Gavrilova et al., 2024; Goodwin et al., 2012; Laurent et al., 2018; Zahn et al., 2001). So far, all axons in *C. elegans* assessed have a plus-end-out microtubule polarity (Goodwin et al., 2012; Harterink et al., 2018; He et al., 2020; Yan et al., 2013). We attempted to determine microtubule polarity in ALA axons by expressing the microtubule plus-end binding protein EBP-2::GFP, but no growing microtubule ends could be detected (data not shown), perhaps because there are few microtubules in the ALA neuron (Ramirez-Suarez et al., 2019) or because they are stable. We therefore assumed a plus-end-out organisation for the assessment of anterograde and retrograde movement from kymographs.

DCV tracks were automatically identified using KymoButler (Jakobs et al., 2019) and then divided into anterograde, retrograde and stationary segments (Table S1, Fig. S2) using our analysis pipeline (Gavrilova et al., 2024). The velocities of the moving segments were plotted as velocity distributions (Fig. S1C–F) and compared to *ida-1::gfp* using two-sample Kolmogorov–Smirnov

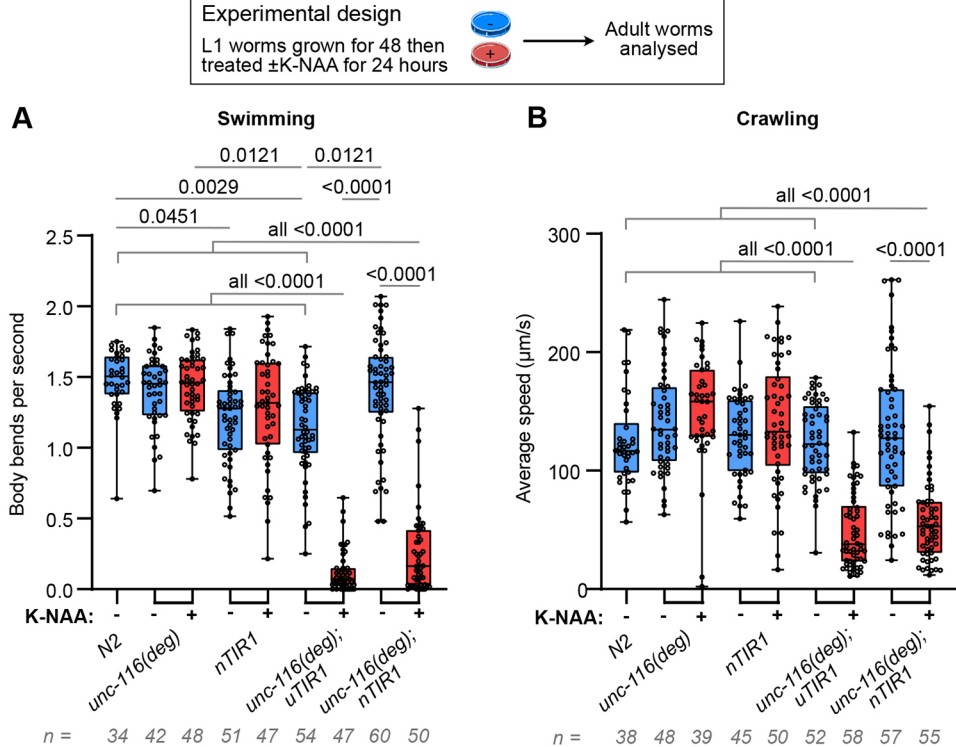

**Fig. 3. Neuronal UNC-116 degradation leads to swimming and crawling defects.** Swimming ability (body bends per second) (A) or crawling ability (average speed) (B) of adult *N2*, *unc-116(deg)*, *nTIR1*, *unc-116(deg);uTIR1* and *unc-116(deg); nTIR1* worms grown on plates with or without K-NAA for 24 h. Graphs represent combined data from three biological repeats, with the number of worms shown below each graph. Boxes display the median and interquartile range with whiskers from the minimum to the maximum value. Only *P*-values ≤0.05 from the Kruskal–Wallis statistical test with Dunn's post-hoc test are shown.

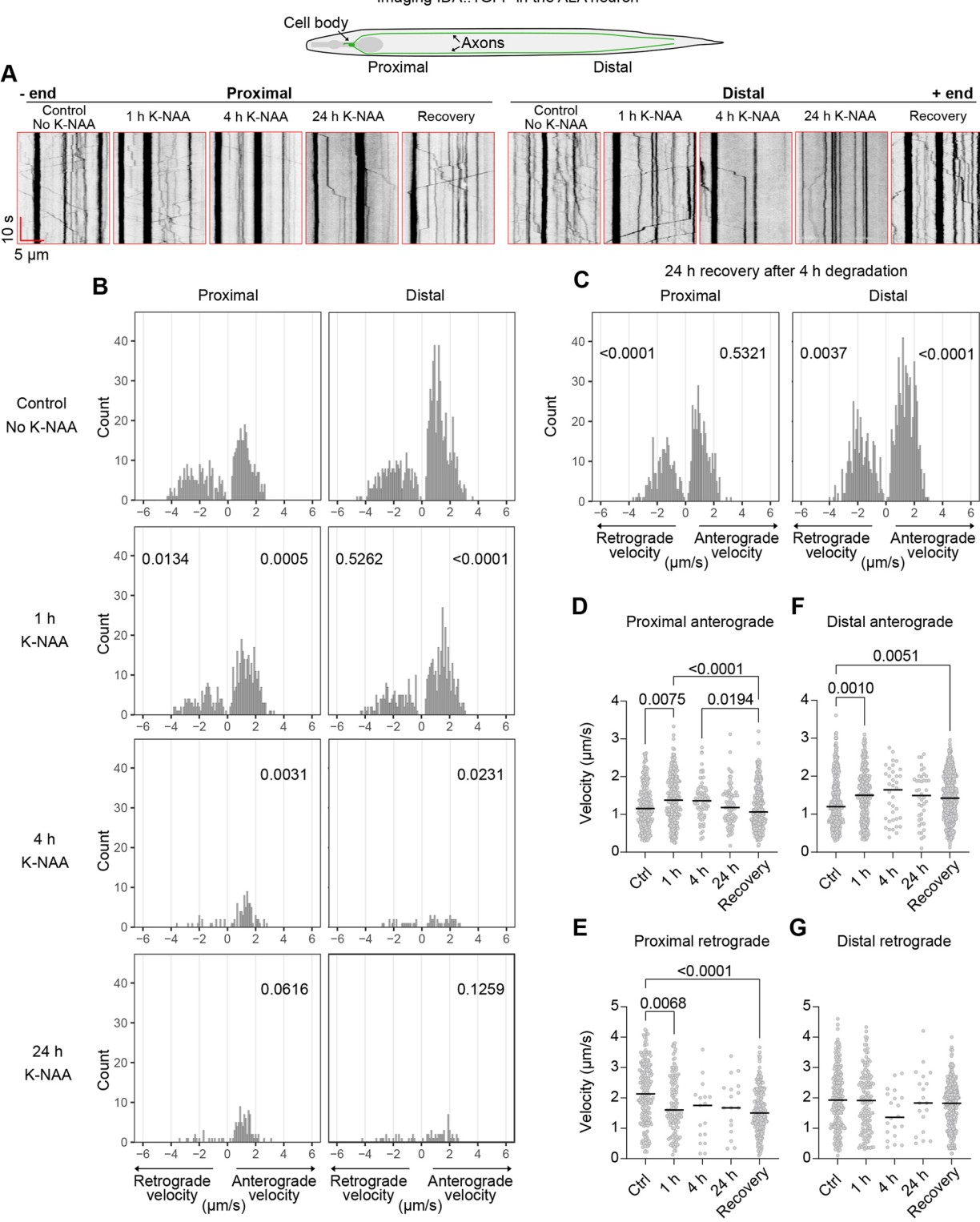

**Fig. 4. Both anterograde and retrograde DCV transport rapidly declines after neuronal UNC-116 degradation and recovers upon K-NAA removal.**
(A) Example kymographs showing DCV movement along the proximal or distal region of the ALA neuron in *unc-116(deg);nTIR1;ida-1::gfp* worms under the indicated conditions. The head is to the left. The diagram shows the ALA neuron in the worm, highlighting the axons imaged. (B) Velocity distributions of moving DCV track segments after incubation without (control) or with K-NAA for 1, 4 or 24 h, with the number of moving segments (count) on the *y*-axis. Between 10 and 15 worms were imaged per strain. The number of kymographs, segments analysed and resulting velocities are given in Table S2. *P*-values from two-sample K-S tests are shown, comparing retrograde and anterograde velocities in each condition and location to the equivalent subset in the untreated worms. The statistical analysis excluded the retrograde data from 4 h and 24 h, due to the limited number of values. (C) Velocity distributions of moving DCV track segments in worms treated with K-NAA for 4 h, then moved to K-NAA-free plates to recover for 24 h. *P*-values from two-sample K-S tests are shown, compared to those from the equivalent subset in the untreated day 1 adults. (D–G) The same data plotted in scatter plots, displaying *P*-values ≤0.05 from Kruskal–Wallis tests followed by Dunn's post-hoc test with lines representing the median for each condition.

(K-S) tests. In addition, Kruskal–Wallis tests were carried out to compare the differences in rank sums between the different groups, used as an indicator of differences in median velocities (Ostertagová et al., 2014) (Fig. S1G–J). Run lengths were not determined as many moving DCVs entered or left the field of view, making it impossible to observe the start and end points of their runs. The presence of the AID*-ALFA tag on UNC-116 had no noticeable effect on the proportion of moving DCVs (Fig. S2A), although there was a slight shift toward lower anterograde velocities in *unc-116(deg); ida-1::gfp* compared to *ida-1::gfp* worms, and the proportion of fast retrograde velocities in the proximal axon increased somewhat (Fig. S1D). The only statistically significant median velocity change was for distal anterograde transport (Fig. S1I). The AID*-ALFA tag therefore had little effect on UNC-116 function.

As background degradation in the absence of K-NAA reduced UNC-116 levels slightly (Fig. 1) and affected swimming (Fig. 2A), we also analysed DCV transport in the AID*-UNC-116 TIR1 strains in the absence of K-NAA. Active DCV transport was seen in both the *unc-116(deg);uTIR1;ida-1::gfp* and *unc-116(deg);nTIR1; ida-1::gpf* strains (Fig. 4; Fig. S1), although the number and percentage of moving DCVs were slightly lower in *unc-116(deg); nTIR1;ida-1::gfp* worms (Table S1, Fig. S2A), and fewer DCVs moved at the highest velocities (Fig. S1E,F). Most median velocities were correspondingly slightly reduced (Fig. S1G–J). Taken together, tagging UNC-116 with an AID*-ALFA tag and the low level of background degradation by TIR1 in the absence of K-NAA caused only minor changes in motility.

### Bidirectional DCV transport is rapidly reduced by UNC-116 degradation

In stark contrast, UNC-116 degradation in the presence of K-NAA rapidly and drastically inhibited DCV motility in *unc-116(deg); nTIR1;ida-1::gfp* (Fig. 4) and *unc-116(deg);uTIR1;ida-1::gfp* (Fig. S3) worms. After only 1 h of degradation, the number of both anterograde and retrograde movements was greatly reduced in the neuronal and ubiquitous degron strains, and further decreased after 4 and 24 h, when moving DCVs were rare (Fig. 4; Figs S2, S3, Movie 5). A few anterograde movements persisted, particularly in the proximal axon, even after 24 h of K-NAA treatment, whereas retrograde motility was almost completely lost. Interestingly, after 1 h of degradation, the median anterograde velocities significantly increased in both the proximal and distal axon in *unc-116(deg); nTIR1;ida-1::gfp* worms, and in the distal region of *unc-116(deg); uTIR1;ida-1::gfp* worms (Fig. 4D,E; Fig. S3F, Tables S2 and S4). In contrast, retrograde velocities were only significantly reduced in the proximal axon in *unc-116(deg);nTIR1;ida-1::gfp* worms after 1 h K-NAA treatment (Fig. 4F,G). Too few retrograde segments were identified at 4 and 24 h to allow statistical comparison of the velocity distributions by a K-S test with the untreated group. Overall, degradation of UNC-116 for as little as 1 h noticeably affects DCV movement in both directions along the length of the ALA neuron, and movement is profoundly stalled after 4 h.

Having assessed motility, we investigated what effect 24 h degradation of UNC-116 in the *unc-116(deg);uTIR1;ida-1::gfp* strain had on the overall distribution of DCVs in the proximal and distal ALA axon. Their distribution per μm and fluorescence intensities were not altered in the proximal axon after K-NAA treatment (Fig. 5A–C; Fig. S4). In contrast, after UNC-116 degradation, DCVs were less frequent along the distal axon shaft but accumulated abnormally at the very tip of the axon, where an intense fluorescence signal was observed (Fig. 5D; Fig. S5). This altered distribution was confirmed quantitatively by comparing the DCV distribution and

signal intensity in the first 80% and final 20% of the distal segment before and after K-NAA (Fig. 5E). Although the number of DCV puncta was reduced throughout the whole distal region after UNC-116 degradation, the IDA-1::GFP intensity of DCVs in the first 80% of the segment was unchanged. In contrast, the mean peak fluorescence intensity greatly increased at the axon tip without UNC-116.

Although the transmembrane protein IDA-1::GFP is an excellent marker for DCV transport in the ALA neuron, we wanted to assess the effects of UNC-116 degradation on a soluble DCV cargo, as well as investigate how global the effect on DCV transport was by imaging a different neuron. Hence, a strain expressing NLP-21:: VENUS (an FMRF amide-related neuropeptide) in a subset of motor neurons (Sieburth et al., 2007) was crossed with the *unc-116(deg);uTIR1* strain. Imaging of NLP-21::VENUS in the commissure of the DB7 neuron (Fig. 6B) revealed active DCV movement that was greatly reduced after UNC-116 degradation for 24 h (Fig. 6A; Movie 6). Interestingly, steady-state analysis of NLP-21::VENUS in the DB7 axon showed that UNC-116 degradation significantly increased fluorescence intensities but reduced the number of puncta per μm (Fig. 6C,D). This observation might indicate that DCVs are less able to release their contents by fusion with the cell membrane when UNC-116 is absent. In case this was due to a general disorganisation of the synaptic region, we assessed the distribution of synapses by crossing a strain expressing the presynaptic marker GFP::SNB-1 (synaptobrevin) under the same promoter (Sieburth et al., 2007) with *unc-116(deg);uTIR1* worms. Synaptic vesicles are transported by UNC-104, so they should be unaffected by UNC-116 degradation (Hall and Hedgecock, 1991; Kumar et al., 2010; Niwa et al., 2016; Ou et al., 2010; Wu et al., 2013; Zheng et al., 2014). Indeed, neither the puncta per μm nor puncta intensity was altered following loss of UNC-116 (Fig. 6E,F). UNC-116 degradation therefore affects DCV transport and possibly delivery without affecting synaptic organisation.

### The effect of K-NAA removal on UNC-116 protein level, DCV motility, and worm locomotion

An open question in neurobiology is whether phenotypes of neurodegeneration can be reversed if a functional transport machinery is restored, or whether transport loss leads to irreversible downstream defects (e.g. d'Ydewalle et al., 2011; Zhu and Sheng, 2011). Therefore, we investigated whether UNC-116 protein levels recovered after K-NAA removal. *unc-116(deg);uTIR1 w*orms were grown on K-NAA plates for 24 h, followed by recovery on fresh NGM plates for different periods of time (Fig. 7A). Some rescue of UNC-116 protein level was detected, but this was statistically insignificant compared to worms treated with K-NAA for 24 h (Fig. 7B). As the rate of recovery of AID-tagged protein could be influenced by the duration of K-NAA exposure, adult worms were left on K-NAA plates for only 3 h, and the recovery experiment was repeated. Protein level recovery was better, but still incomplete (Fig. 7C). UNC-116 was efficiently degraded after a 10 h treatment with K-NAA, with recovery levels intermediate between those seen following the 3 and 24 h degradation (Fig. 7D). UNC-116 protein levels could therefore recover following K-NAA treatment, but the extent was influenced by the duration of original K-NAA exposure.

We next wanted to test whether locomotion would recover after K-NAA removal in the *unc-116(deg);uTIR1* strain used for the biochemical characterisation of UNC-116 rescue (Fig. 7). We initially exposed worms to K-NAA for 24 h and then allowed them to recover without K-NAA for 1, 3, 6, 24 or 48 h (Fig. S6A). Swimming ability did not recover (Fig. S6B), whereas crawling velocity was slightly, but significantly, improved after 24 h without

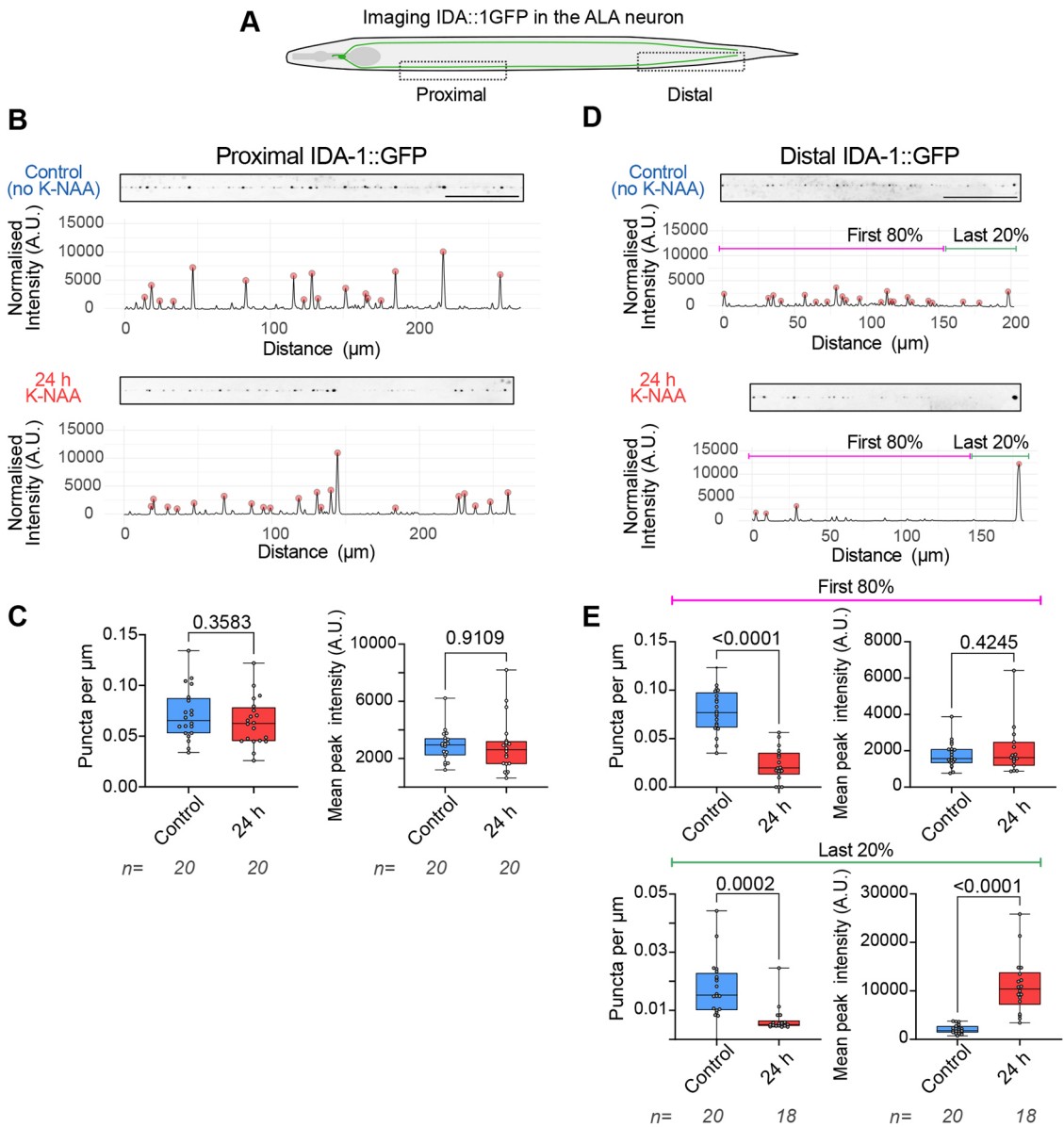

**Fig. 5. Steady-state distribution of DCVs along the ALA axon is altered by UNC-116 degradation.** (A) Distribution of DCVs was analysed just proximal to the vulva (proximal), and at the axon tip (distal). (B,D) Steady-state distribution of DCVs in the proximal (B) and distal (D) ALA axon in untreated (control) worms (top) and worms treated with K-NAA for 24 h (bottom). Scale bar: 50 µm. Intensity profiles from line scans of the presented axon are shown, with the intensity normalised to the median on the y-axis and distance (µm) on the x-axis. (C,E) Density (puncta per µm) and mean puncta (peak) intensity for each worm analysed. For analysis of the distal segment, the intensity profiles were split into the first 80% and last 20% of the total distance (E). Number of worms analysed is indicated below the graphs. Boxes display the median and interquartile range with whiskers from the minimum to the maximum value. *P*-values from unpaired two-tailed *t*-tests are shown. Intensity profiles from all worms analysed are presented in Fig. S4 (proximal) and Fig. S5 (distal). A.U., arbitrary units.

K-NAA (Fig. S6C). After 48 h, crawling ability worsened again, possibly due to ageing effects.

Because the degree of rescue of UNC-116 protein was affected by the duration of initial K-NAA treatment, we investigated the effects on locomotion of reversing a 10 h treatment, which gave a partial rescue of UNC-116 protein level after 24 and 48 h without K-NAA (Fig. 7D). Importantly, a 10 h K-NAA treatment reduced adult crawling speed (Fig. 2G) with only a minor, statistically insignificant, effect on swimming (Fig. 2I). Although the 10 h K-NAA treatment did not affect swimming ability when assessed immediately after treatment, to our surprise we found that swimming was strongly inhibited after a further 24 or 48 h incubation without K-NAA

(Fig. 8B). In contrast, crawling speed was significantly reduced after K-NAA treatment but recovered back to normal levels after 24 h without K-NAA (Fig. 8C). There was also a rescue in crawling speed after 48 h, but worms crawled more slowly than at 24 h, possibly due to ageing effects.

We then tested how DCV motility in the ALA neuron was affected by 10 h in K-NAA, followed by a 24 h rescue in the neuronally-expressed TIR1 strain (Fig. 8D–H; Table S3). As expected, the number of DCV movements in both directions was greatly reduced after a 10 h K-NAA treatment (Fig. 8D). The median anterograde and retrograde velocities of DCVs in K-NAA-treated worms were significantly lower than those of controls in the proximal axon but

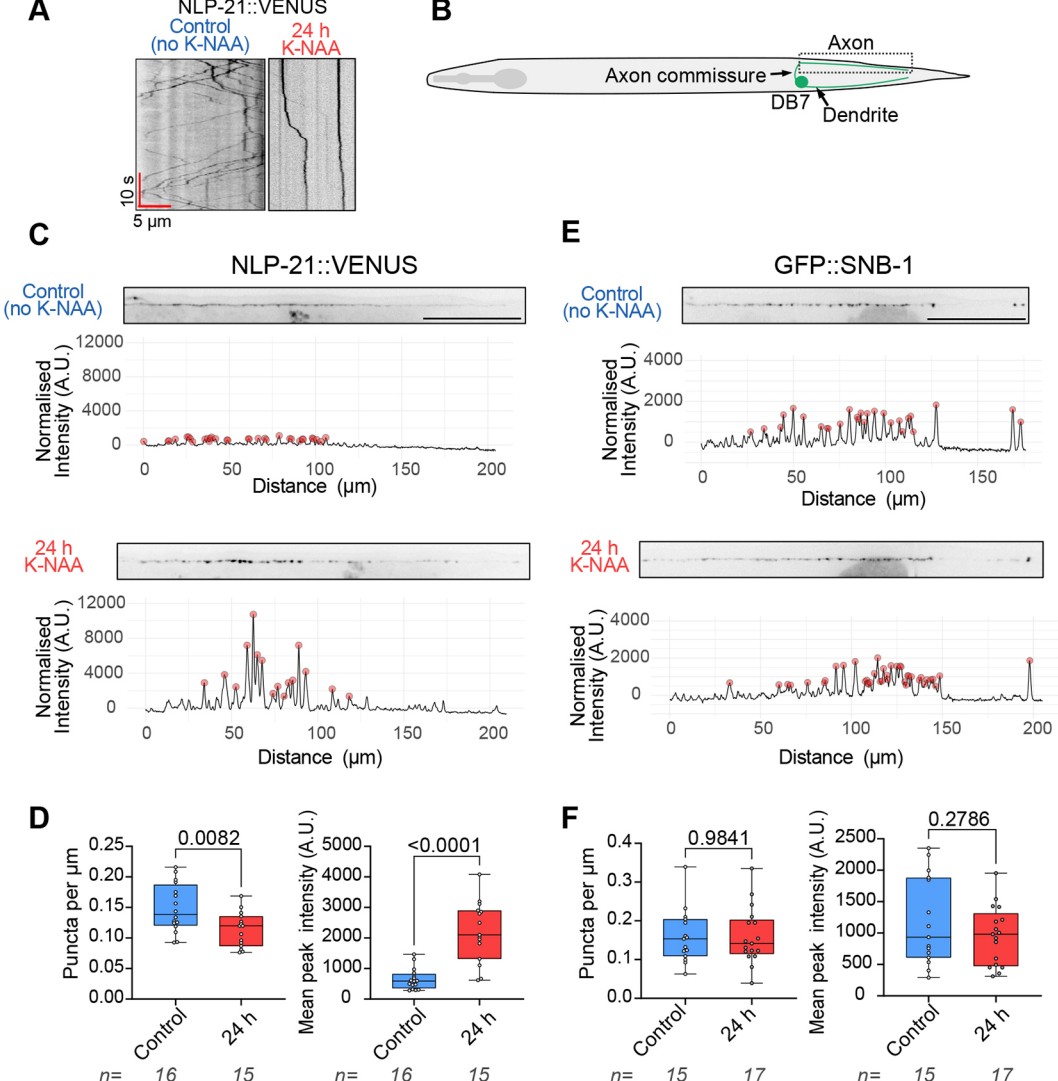

**Fig. 6. DCV distribution and motility is altered by UNC-116 degradation in the DB7 motor neuron, but synaptic organisation is unaffected.** (A,B) Movement of DCVs, visualised by NLP-21::VENUS, in the axon commissure of the DB7 motor neuron (diagram in B) in untreated (control) worms and worms treated with K-NAA for 24 h. (C,E) Steady-state distribution of DCVs (C) and synaptic vesicles (GFP::SNB-1) (E) in the DB7 axon in untreated (control) worms (top) and worms treated with K-NAA for 24 h (bottom). Scale bars: 50 μm. Intensity profiles from line scans of the presented axon are shown, with the intensity normalised to the median on the *y*-axis and distance (μm) on the *x*-axis. (D,F) Density (puncta per μm) and mean puncta (peak) intensity for each worm analysed. Boxes display the median and interquartile range with whiskers from the minimum to the maximum value. Number of worms analysed is indicated below the graphs. *P*-values from unpaired two-tailed *t*-tests are shown. A.U., arbitrary units.

not the distal axon (Fig. 8E–H; Table S3). Following 24 h without K-NAA, a partial recovery of the number of DCV movements in both directions was seen, but the median velocities in all mutant strains remained lower than for controls (Fig. 8D–H).

As UNC-116 protein levels recovered better after shorter K-NAA treatments in *unc-116(deg);uTIR1* worms (Fig. 7) and DCV motility was already strongly inhibited in both *unc-116(deg); uTIR1;ida-1::gfp* and *unc-116(deg);nTIR1;ida-1::gfp* worms after a 4 h incubation with K-NAA (Fig. 4B, Fig. S3A), we tested whether DCV transport recovered without K-NAA for 24 h. Indeed, the number of DCV movements in both directions increased strongly, with the best recovery seen in the distal region (Fig. 4A,C; Fig. S2B, Table S2). Interestingly, the anterograde velocities returned to control levels in the proximal axon but were slightly faster than controls in the distal axon (Fig. 4D,F). In contrast, retrograde velocities also recovered fully in the distal region but were slower in the proximal region (Fig. 4E,G).

Altogether, these data show that removal of K-NAA after a 10 h treatment can lead to partial recovery of UNC-116 protein levels and DCV motility, and that this is sufficient for rescue of crawling ability. However, under the same conditions, swimming is initially unaffected but then deteriorates during the rescue incubation, suggesting that swimming is particularly sensitive to long-term reductions in UNC-116 function.

## DISCUSSION

AID offers exciting opportunities for investigating the function of essential proteins by controlled degradation. We have harnessed this approach to study the role of kinesin-1 in worms at specific developmental stages, exploring two levels of biological complexity – DCV motility and worm locomotion. We find that UNC-116 is degraded rapidly, and that bidirectional DCV movement is inhibited on a similar timescale. In contrast, it takes much longer for worm locomotion to be affected.

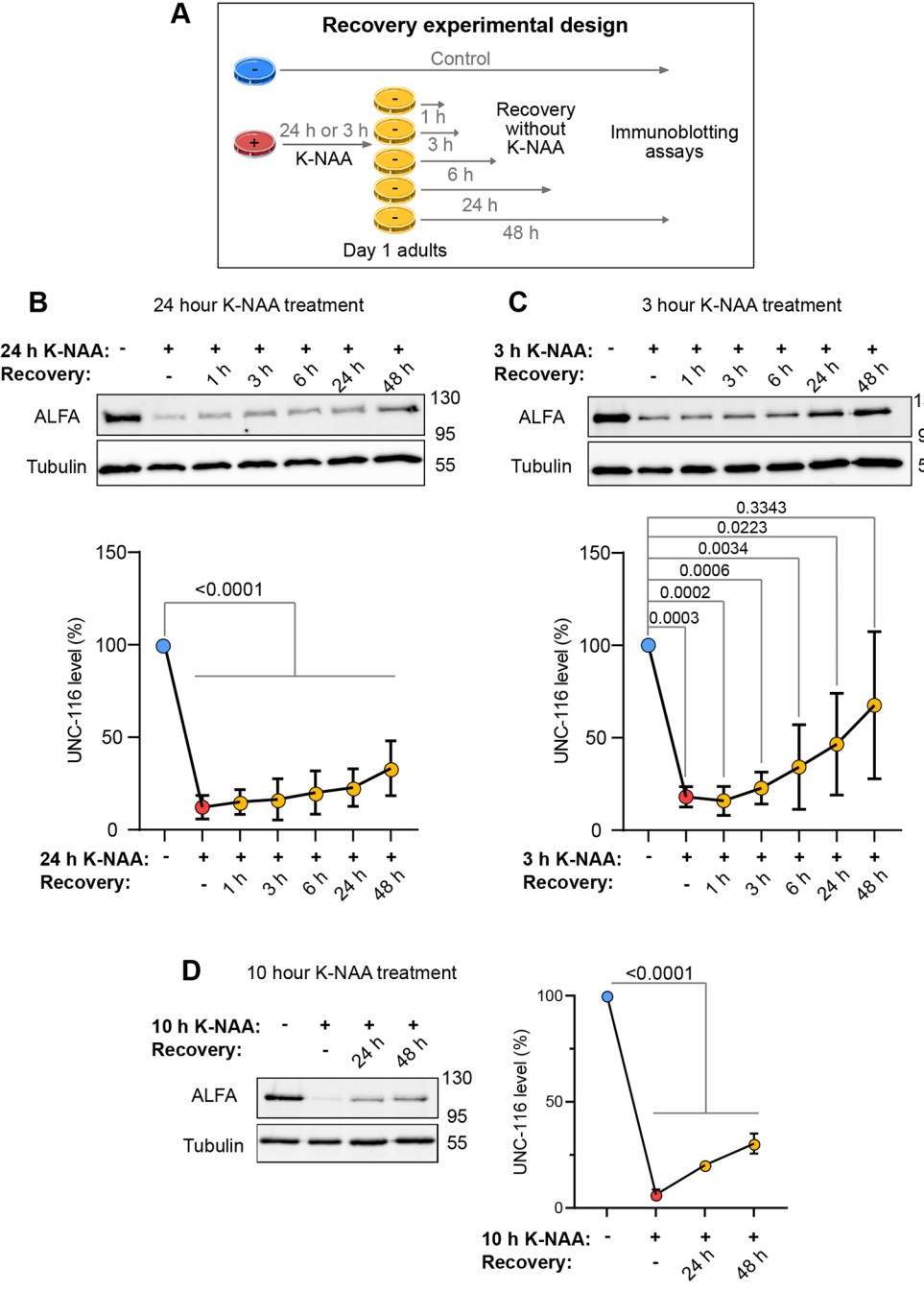

**Fig. 7. Recovery of UNC-116 protein levels after removal of K-NAA.** (A) Experimental design to test for UNC-116 protein recovery. (B–D) *unc-116(deg);uTIR1* adults were treated with K-NAA for 24 h (B), 3 h (C) or 10 h (D), followed by rescue for the time indicated, with protein levels assessed using anti-ALFA antibodies. Quantification of ALFA-AID*-UNC-116 protein is shown, normalised relative to anti-tubulin and the control ALFA signal (ctrl), which was from adults age-matched to the experimental endpoint. Means±s.d. (*N*=3 independent experiments for B, *N*=4 for C, *N*=3 for D) and *P*-values from one-way ANOVA followed by post-hoc Dunnett test are shown.

## The role of kinesin-1 in worm locomotion

Studies that assess phenotypes in adult *unc-116* mutants face the problem that kinesin-1 plays important roles in *C. elegans* development, meaning that it can be difficult to interpret the root cause of any defects seen. Neuronal development begins during *C. elegans* embryogenesis and continues at the L1 and L2 stages (Altun and Hall, 2011; Hedgecock et al., 1987; Sulston et al., 1983). By the L4 stage, the expression of genes involved in neurite development drops (Godini et al., 2022). UNC-116 is essential for organising dendritic microtubules during early development, which is in turn crucial for trafficking of cargoes and organelles (Harterink et al., 2018; He et al., 2020; Yan et al., 2013). UNC-116 is also important for the transport of factors involved in axonal outgrowth (Drozd and Quinn, 2023; Lai and Garriga, 2004; Sakamoto et al., 2005; Su et al., 2006). Microtubule sliding

mediated by kinesin-1 is crucial for establishing initial neurite polarity and outgrowth, after which the microtubule cytoskeleton is largely immobilised (He et al., 2020; Lu et al., 2013).

Using AID, we degraded UNC-116 in L4 and adulthood, when the nervous system is fully developed and axonal-dendritic microtubule polarity established. This generates a similar uncoordinated phenotype whether UNC-116 is degraded in all cells or just neurons (Fig. 3). This loss of mobility is likely due to failure in delivery of neuronal cargoes needed to maintain locomotory circuit function. Synaptic vesicle delivery should not be affected, however, as they are transported by the kinesin-3 protein UNC-104 (Hall and Hedgecock, 1991; Kumar et al., 2010; Niwa et al., 2016; Ou et al., 2010; Wu et al., 2013; Zheng et al., 2014), and we see no noticeable change in the distribution of SNB-1 at synapses in DB7 neurons (Fig. 6). Instead, the locomotion defects seen when UNC-116 is lost are likely due to

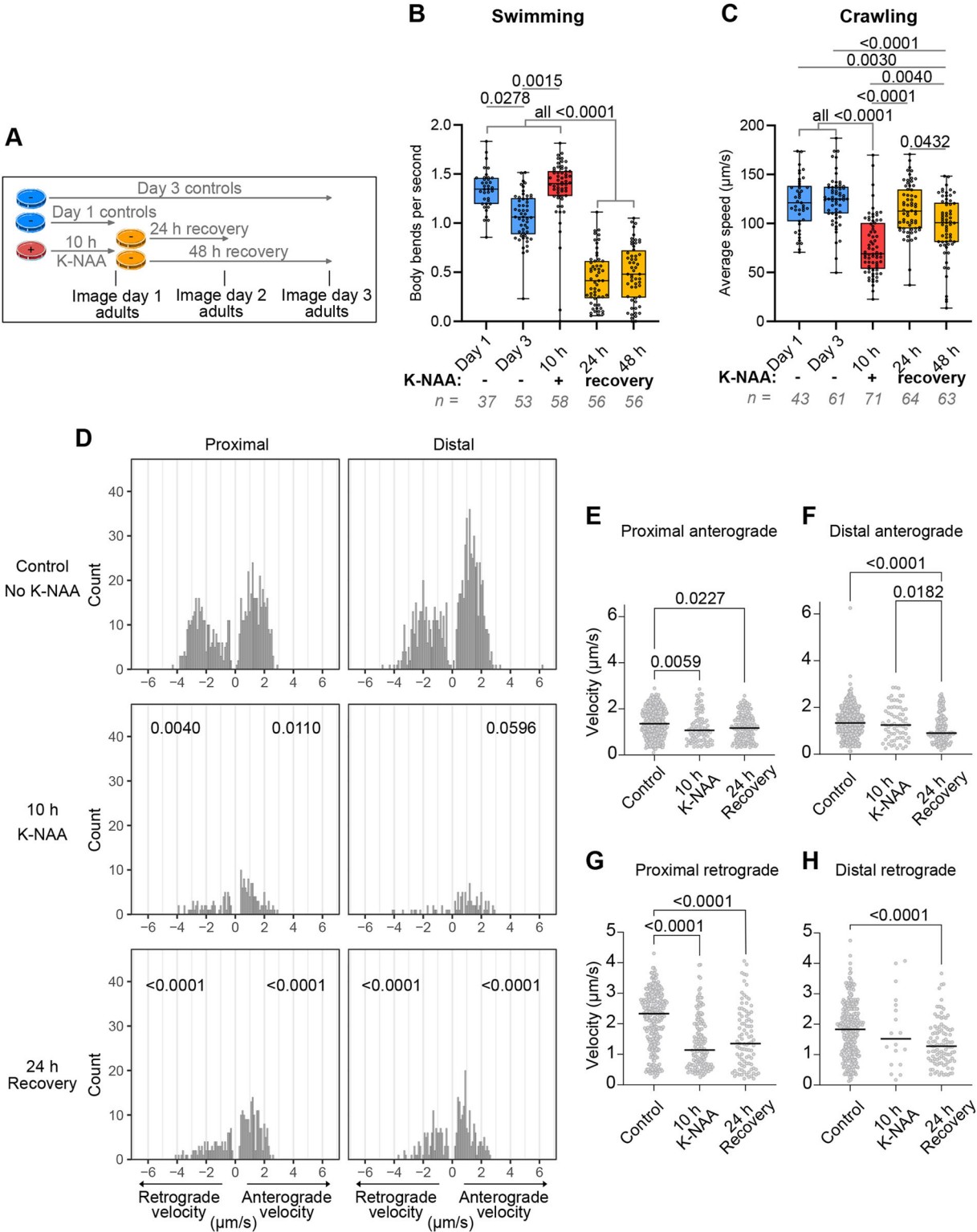

**Fig. 8. Crawling, swimming and DCV motility respond differently to the removal of K-NAA.** (A–C) Time-course analysis (A) of swimming (B) and crawling (C) in *unc-116(deg);uTIR1* adults treated with K-NAA for 10 h then transferred to plates without K-NAA for the duration indicated. The number of worms analysed across three independent experiments is indicated below each graph, with boxes displaying the median and interquartile range with whiskers showing the data range. *P*-values ≤0.05 from the Kruskal–Wallis statistical test with Dunn's post-hoc test are shown. (D) Velocity distributions of moving DCV track segments in *unc-116(deg);nTIR1;ida-1::gfp* worms incubated without (control) or with K-NAA for 10 h, followed by rescue from K-NAA for 24 h, with the number of moving segments (count) on the *y*-axis. Between 10 and 12 worms were imaged per strain. The number of kymographs, segments analysed and resulting velocities are given in Table S3. *P*-values are shown from two-sample K-S tests comparing retrograde and anterograde velocities in each condition and location to the equivalent subset in untreated worms. The statistical analysis excluded the distal retrograde data from 10 h K-NAA due to the limited number of values. (E–H) The same data plotted in scatter plots displaying *P*-values from Kruskal–Wallis tests followed by Dunn's post-hoc test with lines representing the median for each condition.

disrupted trafficking of cargos including post-synaptic receptors and DCVs.

Worm locomotion is principally controlled by cholinergic motoneurons that trigger muscle contraction on one side of the worm while GABAergic motoneurons cause simultaneous muscle relaxation on the opposite side (Cohen and Sanders, 2014; Haspel et al., 2020; Zhen and Samuel, 2015). This coordination relies on inputs from a number of mechanosensory and command interneurons controlling forward and backward motion (Gjorgjieva et al., 2014). Studies in mice have revealed that kinesin-1 is involved in the transport of GABA$_A$ receptors on vesicular cargo (Nakajima et al., 2012; Twelvetrees et al., 2010), as well as dopamine receptors (Cromberg et al., 2019). In worms, cholinergic motoneurons are activated by command interneurons, such as the AVA, which respond to glutamate (Haspel et al., 2020; Hoerndli et al., 2013, 2015; Von Stetina et al., 2005). UNC-116 in a complex with KLC-2 transports vesicles containing AMPA-type glutamate receptor (AMPAR) subunits to postsynaptic sites in the AVA (Hoerndli et al., 2013, 2015). AMPAR removal from the post-synaptic region by endocytosis might also depend on kinesin-1 function (Brachet et al., 2021; Hoerndli et al., 2013). Thus, defective AMPAR trafficking in command interneurons that have lost UNC-116 might be one explanation for the gradual loss of locomotion. Kinesin function might also be important in motoneurons themselves, as AID of UNC-116 in the DA9 cholinergic motoneuron has been stated to lead to uncoordinated movement, although the phenotype was not analysed (Glomb et al., 2023).

Our data demonstrate the importance of UNC-116 for DCV dynamics. The neuropeptides delivered by DCVs are crucial for controlling worm locomotion (Cai et al., 2004; Charlie et al., 2006; Edwards et al., 2009; Flavell et al., 2013; Hu et al., 2011; Hums et al., 2016; Oranth et al., 2018) with specific roles in proprioception (Ji et al., 2023; Tao et al., 2019) and mechanosensing (Hu et al., 2011). Of the 200 neuropeptides encoded in the worm genome, 76 are expressed in motoneurons (Smith et al., 2024), with each neuron class expressing a distinct combination of neuropeptides and receptors (Taylor et al., 2021). The neuropeptide connectome is extraordinarily complex, allowing individual neurons to signal to many others (Ripoll-Sánchez et al., 2023), with at least 53 pairs of neurons communicating only through DCVs, rather than via electrical or chemical synapses (Randi et al., 2023).

Although DCV signalling between neurons can be sub-second (Randi et al., 2023), DCVs cannot be refilled, unlike synaptic vesicles. The effects of impaired delivery of DCVs to release sites after UNC-116 degradation, and the possible block in cargo release (Fig. 6), could therefore contribute substantially to the locomotion defects seen and would become more severe over time. Likewise, the effect of disrupted trafficking of post-synaptic receptors (which will turn over quite slowly) would take time to manifest. This likely explains why the disruption of swimming and crawling takes considerably longer than the loss of cargo motility after UNC-116 degradation.

### Differential effects of UNC-116 degradation on swimming and crawling

Interestingly, we also saw differences in the response of crawling versus swimming to UNC-116 degradation. There is controversy over whether crawling and swimming are two distinct gaits (Deng et al., 2021; Hassinan et al., 2024; Pierce-Shimomura et al., 2008; Vidal-Gadea et al., 2011) or the extremes of one single gait (Berri et al., 2009; Boyle et al., 2012; Butler et al., 2015; Fang-Yen et al., 2010; Korta et al., 2007; Wen et al., 2012). We found that swimming ability was affected faster in L4s than in adults, whereas crawling ability was

impaired sooner in adults than L4 worms (Fig. 2). In addition, although K-NAA removal following a 10 h treatment led to the rescue of crawling (Fig. 8C), swimming ability was initially unaffected but then became much worse during further incubation without K-NAA (Fig. 8B). These data fit better with the two-gait model.

The distinct effects of UNC-116 degradation on L4 and adult swimming and crawling could relate to differences in the availability of neurotransmitter and receptor reserves at neuromuscular junctions and synapses at different life stages. The switch from crawling to swimming is influenced by serotonin signalling, whereas the transition from swimming to crawling is controlled by dopamine (Pierce-Shimomura et al., 2008; Vidal-Gadea et al., 2011). Interestingly, some components of the serotonin signalling pathway are more highly expressed in adults than in L4 worms, whereas the reverse is true for dopamine signalling components (Boeck et al., 2016). Whether UNC-116 loss affects these pathways differentially remains to be tested.

### Kinesin-1 and the regulation of dense core vesicle transport

This work shows conclusively that the dependence on UNC-116 for efficient bidirectional transport of DCVs (Gavrilova et al., 2024) is unrelated to any developmental defects in the hypomorphic mutants used in that study. Thus, when the *C. elegans* nervous system has been allowed to develop normally, the post-developmental loss of the UNC-116 protein rapidly leads to the disruption of bidirectional DCV transport.

Different motors often colocalise on cargoes, sometimes binding via the same adaptor proteins that regulate unidirectional transport by selective motor activation (e.g. Canty et al., 2023; Cason and Holzbaur, 2023; Fenton et al., 2021; Kendrick et al., 2019). Indeed, studies have reported the co-transport of opposing motors on DCVs. For example, in rat hippocampal neurons, the kinesin-3 family member KIF1A remains associated with DCVs during retrograde transport (Lo et al., 2011). Moreover, in *Drosophila melanogaster* neurons, kinesin-1 and kinesin-3 colocalise on DCVs (Lim et al., 2017), a phenomenon also seen in Rab6-positive secretory vesicles in mammalian cells, which are similar to neuronal DCVs (Serra-Marques et al., 2020). These findings indicate that all three opposing motors may be bound to DCVs at the same time.

If this is the case, one might expect dynein-mediated transport of DCVs to continue after degradation of UNC-116. Our steady-state images of the distal ALA neuron suggest that this might occur to a small degree, as the number of DCVs in the distal axon decreases greatly (Fig. 5; Fig. S5). However, there is no change in DCV distribution in the proximal axon (Fig. 5B,C; Fig. S4), and dynein-driven DCV motility is rapidly and profoundly inhibited after UNC-116 degradation (Fig. 4; Fig. S3). This effect mirrors what is seen in a hypomorphic *unc-116* mutant, where both directions of transport are inhibited (Gavrilova et al., 2024).

How might loss of kinesin-1 lead to decreased dynein motility? One possibility is that a shared adaptor that binds both kinesin-1 and dynein is responsible for co-ordinating the direction of movement, but only when both motors are present. The specific adaptor(s) mediating this process is unclear, although BICD is a candidate, as its isoforms have been found to associate with different motors on cargoes, including DCVs and Rab6-positive vesicles (Ali et al., 2025; Grigoriev et al., 2007; Hoogenraad et al., 2001; Matanis et al., 2002; Schlager et al., 2010, 2014). Recent work in *Drosophila* has shown that the small GTPase Rab2 controls DCV movement (Lund et al., 2021) and that a network of Rab2, JIP3 and JIP4, the novel dynein adaptor RUFY and Arl8 provides a means of recruiting kinesin-1, kinesin-3 and dynein (Lund et al., 2026). However, as dynein and its associated complex dynactin are transported

anterogradely separately on different cargoes in cultured human i3 neurons (Fellows et al., 2024), it is also possible that a transport-ready dynein–dynactin complex is not present on kinesin-1-driven DCVs, which therefore cannot move retrogradely when UNC-116 is degraded. In that case, a prediction would be that DCVs with bound dynein or dynactin at the start of K-NAA treatment would continue moving retrogradely, but this pool might be quickly depleted from the axon. Imaging at earlier timepoints after K-NAA addition might help distinguish between these possibilities.

Our results, and those we reported recently (Gavrilova et al., 2024), show that the kinesin-1 UNC-116 is an important anterograde DCV motor in *C. elegans*. UNC-104 also clearly plays a role in DCV motility (Barkus et al., 2008; Goodwin et al., 2012; Lo et al., 2011; Sieburth et al., 2007; Zahn et al., 2004), primarily by transporting DCVs out of the cell body and through the initial axon (Gumy et al., 2017; Lim et al., 2017; Park et al., 2023). Importantly, we find that some anterograde DCV movement persists at later times after UNC-116 degradation, particularly in the proximal axon (Fig. 4; Figs S2, S3), and that DCVs accumulate at the distal tip of the ALA axon (Fig. 5D). Furthermore, after 1 h in K-NAA, anterograde velocities were significantly increased (Fig. 4). These observations indicate that the faster motor UNC-104 (Kita et al., 2024) might be able to transport DCVs when some UNC-116 protein remains, but that robust UNC-116 degradation leads to an overall loss of motility. Curiously, in the recovery condition, anterograde velocity in the distal axon (Fig. 4F) remained higher than in the control. Altogether, our data indicate that although the kinesin-3 UNC-104 is active throughout the length of the axon in *C. elegans*, UNC-116 is the dominant anterograde motor once DCVs leave the cell body.

Clearly, an intricate balance between motors is essential, and potentially a mechanism by which they regulate each other. Previously, kinesin-1 but not kinesin-3 was found to engage in tug-of-war with dynein (Serra-Marques et al., 2020), and kinesin-3 detaches more easily from microtubules than kinesin-1 (Arpağ et al., 2014, 2019; Norris et al., 2014). Thus, association with microtubules by UNC-116 might be required for UNC-104 on DCVs to efficiently bind to microtubules and transport the vesicle, a model which has been previously suggested (Hancock, 2014). Importantly, because we are degrading tagged wild-type protein, rather than using partial loss-of-function mutants, we can rule out dominant-negative effects on bidirectional transport caused by an aberrant motor.

## Advantages and disadvantages of the UNC-116-AID system

The UNC-116 AID system avoids the use of mutants that have developed aberrantly, allowing UNC-116 to be removed from worms at any stage of development. In combination with tissue-specific expression of TIR1, kinesin-1 can be degraded only in specific cell types. Here, we find that the consequences of neuronal UNC-116 degradation on DCV motility and worm locomotion mirror those of ubiquitous degradation, suggesting that non-neuronal UNC-116 plays only a minor role in worm swimming and crawling. Future studies targeting UNC-116 in specific neurons will allow further dissection of the importance of kinesin-1 for locomotion.

Although the current UNC-116 AID system is a powerful tool, a limitation is the somewhat leaky degradation seen in the *unc-116(deg);uTIR1* strain in the absence of added K-NAA. Basal degradation of AID-fused proteins by TIR1 is proposed to be due to non-specific binding of TIR1 to the AID protein, or to the production of auxin-like indoles by commensal bacteria in the worm gut (Hills-Muckey et al., 2022; Kanke et al., 2011; Natsume et al., 2016; Negishi et al., 2022; Schiksnis et al., 2020). In *unc-116(deg);uTIR1*, some basal degradation was detected at the protein level, slightly

affecting swimming ability (Fig. 1; Fig. 2A) and the velocity of DCV transport (Fig. S1). Using the TIR1$^{F79G}$ mutant that both improves the specificity of ligand binding and functions at very low concentrations of synthetic 5′Ph-auxin (Hills-Muckey et al., 2022; Negishi et al., 2022) might improve the specificity of UNC-116 degradation. Given that recovery from auxin treatment can be concentration dependent (Zhang et al., 2015), the low concentration of 5′Ph-auxin used with TIR1$^{F79G}$ could be an advantage for future studies.

We have shown that conditional degradation of a microtubule motor can be used to study the functions of this motor in depth, including dissecting unknown aspects of cargo transport and the systemic effects of motor loss. Extending this work to include AID of dynein and UNC-104 will allow the rapid manipulation of each DCV motor, providing a testbed for unravelling how UNC-116, UNC-104 and dynein work together, without resorting to the use of mutants with partially functional motors.

## MATERIALS AND METHODS

### C. elegans culture, strains and genetics

*C. elegans* strains were cultured on nematode growth medium (NGM) plates seeded with the *Escherichia coli* strain OP50 at 20°C, as previously described (Stiernagle, 2006). All strains used in this study are listed in Table S5. The OL332 and OL370 strains generated in this study will be made available via the *Caenorhabditis* Genetics Center (CGC). Other strains generated are available from the authors upon request.

The *unc-116(deg)* strain was generated by InVivo Biosystems using CRISPR-mediated insertion. Two sgRNAs targeting the 5′ end of the *unc-116* open reading frame were used to guide CRISPR/Cas9 (sgRNA1: 5′-AAAATGGAGCCGCGGACAGA-3′+PAM and sgRNA2: 5′-CGGACAGACGGAGCAGAATG-3′+PAM). A single-stranded oligodeoxynucleotide (ssODN) donor homology repair template was generated, containing the DNA sequence encoding AID*-ALFA and flanking homology arms (Fig. S7). The sgRNAs were complexed with Cas9 protein prior to injection, followed by microinjection of the complete CRISPR/Cas9 gene editing mix into the gonads of young adult hermaphrodites. F1 animals were screened for the presence of co-CRISPR phenotype [*dpy-10 (cn64)*, chromosome II]. Homozygotes were confirmed by PCR and Sanger sequencing and backcrossed three times to *N2*.

The *nTIR1* extrachromosomal array strain was generated by InVivo Biosystems by microinjection of a mix of 15 ng/µl pNU3232 (*Prab-3::3xFLAG::TIR1::tbb-2 3′UTR*), 2 ng/µl pNU3225 (*Pmyo-2::NLS::GFP::unc-54 3′UTR*), 15 ng/µl pNU936 (*Punc-119::unc-119::unc-119 3′UTR*) and 69 ng/µl salmon testes DNA into *unc-119 (ed3)* null worms, followed by selection of worms with GFP expression and phenotypic rescue.

The OL0495 (*ida-1p::ebp-2::wrmScarlet*) strain was engineered using transposon-based Mos1-mediated Single Copy Insertion method (Frøkjær-Jensen et al., 2008). The transgene was cloned into pCFJ150 (Addgene plasmid #19329) by Gateway assembly, generating the plasmid pQ050 (*ida-1p::ebp-2::wrmScarlet*). A mix of pQ050, transposase plasmid pCFJ601 (Addgene plasmid #34874), peel-1 negative selection plasmid pMA122 (Addgene plasmid #34873) and three fluorescent markers for negative selection [pCFJ90 (Pmyo-2::mCherry, Addgene plasmid #19327), pCFJ104 (Pmyo-3::mCherry, Addgene plasmid #19328), and pGH8 (Prab-3::mCherry, Addgene plasmid #19359)] was injected into EG8082 for insertion on chromosome V *(oxTi365)*. Moving, nonfluorescent worms were selected, and insertions were confirmed by PCR genotyping.

### Genotyping by single-worm PCR

DNA was amplified using OneTaq Quick-load 2X Master Mix (M0486L, New England Biolabs) according to the manufacturer's protocol. Edit primer pairs (5′-CAGGATCTACATCTGGATCACC-3′, 5′-GAGAGA-ATTGACACACCTGC-3′) were used to verify that worms contained the AID*-ALFA CRISPR insert. Flanking primer pairs (5′-ATTGCAGG-CATTGTAAGGAGAAGC-3′, 5′-CATGTGAGAGAATTGACACACCT-GC-3′) were used during crosses to confirm homozygotes for the CRISPR edit.

## Synchronisation by bleaching

Plates with high densities of gravid adult hermaphrodites were washed with M9 buffer (22 mM $KH_2PO_4$, 86 mM NaCl, 42 mM $Na_2HPO_4$ and 1 mM $MgSO_4$) followed by aspiration into 15 ml tubes. Worms were pelleted by centrifugation at 500 $g$ for 1 min, followed by aspiration of most of the supernatant. 1 ml of bleaching solution [0.7 M NaOH and 20% (v/v) sodium hypochlorite solution in $ddH_2O$] was added to the pellet and vortexed periodically for 5 min, ensuring disintegration of the mothers but not the eggs. Three M9 washes were carried out, pelleting the released eggs by centrifugation (1000 $g$ for 1 min) between each wash. After the final wash, 5 ml of M9 solution was added, and tubes were left at room temperature on a rocker for at least 24 h, ensuring hatching and arrest at the L1 stage. For experiments at the L2 stage, L1 worms were pelleted (500 $g$ for 1 min), aspirated and allowed to grow on plates for 24 h. For experiments at the L4 stage, L1s were grown for 48 h, and for the adult stage, L1s were grown for 72 h.

## K-NAA treatment

Potassium 1-naphthaleneacetate (K-NAA) (N0006, TCI chemicals) was diluted in $ddH_2O$ and kept as a 100 mM stock at 4°C. 1 mM K-NAA plates were prepared as previously described (Martinez and Matus, 2020). Because auxin inhibits the growth of OP50 (Zhang et al., 2015), a 50 µl drop of 10× concentrated OP50 was added to each plate and allowed to dry at room temperature on the day of use. Synchronised worms were added to K-NAA plates for variable durations depending on the experimental procedure.

## Generation of worm lysates and western blotting

Worms were transferred to fresh NGM plates without OP50. 100 L2 worms/condition, 30 L4 worms/condition, or 20 adult worms/condition were picked except in experiments with 72 h degradation (Fig. 1C), where 25 K-NAA-treated *unc-116(deg);uTIR1* and *unc-116(deg);nTIR1* adults were transferred due to their smaller size, in order to obtain approximately equal amounts of protein in each sample. Worms were washed off with M9 and added to Eppendorf tubes, followed by centrifugation at 500 $g$ for 1 min and aspiration of the supernatant, leaving ∼20 µl with the worm pellet. 20 µl of 2× sample buffer was added to the tubes, followed by boiling at 95°C for 10 min and SDS-PAGE. The gel was transferred onto a 0.45 µm pore PVDF membrane (IPFL00010, Millipore) using a Bio-Rad wet transfer system. Antibodies were made up in TBS-T (20 mM Tris-HCl, pH 7.6-7.8, 150 mM NaCl and 0.1% Tween-20) according to the dilutions in Table S6. Washed membranes were visualised using a LI-COR Odyssey Fc scanner. Band intensities were quantified using the Fiji Gel Analyzer function. Uncropped images of blots from the work are shown in Fig. S8.

## Crawling and swimming assays

Movies of crawling and swimming behaviour were captured using a Leica M165 FC Fluorescent Stereo Microscope with an optiMOS Scientific CMOS Camera and PLANAPO 1.0× objective. Assays were based on protocols from Bayrakli et al. (2015) and Hahm et al. (2015). For crawling assays, 10–20 worms were placed on a fresh NGM plate without OP50. For swimming assays, worms were added to fresh NGM plates, followed by the addition of M9 buffer. Image streams of 500 frames at ∼16.6 frames per second were captured using Micro-Manager. Movies were analysed using the Fiji plugin wrMTrck (Nussbaum-Krammer et al., 2015) to quantify the number of body bends per second and average crawling speed, calculated as the sum of the length of all movement vectors for each track (µm) divided by the time recorded (s). Each measurement represents one worm. Three independent repeats were performed for each assay, and the data from multiple movies was combined for each strain or condition. During time-course analyses, time points were staggered and analysed within ∼1 h of each other.

## Confocal imaging

For time-course experiments, time points were staggered and imaged within 3–4 h of each other in day 1 adults. The recovery condition was imaged a day later. To mount worms for imaging, 2% agarose pads were generated by placing a drop of melted agarose on a vinyl record and then flattening it with a glass slide, as outlined previously (Rivera Gomez and Schvarzstein, 2018).

Worms were immobilised in 10 mM tetramisole hydrochloride (T1512, Sigma-Aldrich) diluted in M9 and placed in the resulting grooves in the solidified agarose, ensuring that they were parallel to the image view, and then covered with an 18 mm coverslip. The edges of the coverslip were sealed with VALAP (a 1:1:1 mixture of vaseline, lanolin and paraffin).

Confocal images were acquired using a CSU-X1 spinning disc confocal (Yokagawa) on a Zeiss Axio-Observer Z1 inverted microscope with a 100×/1.30 (2D timelapses) or 40×/1.30 (3D steady-state images) Plan-Apochromat objective, a Prime 95B Scientific CMOS (1200 x1200 11 µm pixels; backlit; 16-bit) camera (Photometrics) and a motorised *XY* stage (ASI) with a piezo Z insert (Mad City Labs), using Slidebook 2023 (3i) software. The 488 nm laser was controlled using an AOTF through the LaserStack [Intelligent Imaging Innovations (3i)].

## Kymograph analysis

Continuously streamed images of a single *Z*-plane were captured for 500 frames with a frame rate of 156 or 184 ms. Proximal movies were captured up to 300 µm from the cell body, and distal movies up to 300 µm from the axon tip.

Raw TIFF files were stabilised using the image stabiliser plugin (https://www.cs.cmu.edu/~kangli/code/Image_Stabilizer.html) for Fiji, using the best suitable initial reference slice. Standard parameters for the image stabiliser were used. A segmented line was then superimposed over the axon, and kymographs were generated using the multi kymograph plugin (https://biii.eu/multi-kymograph). Tracks were identified using KymoButler (Jakobs et al., 2019). Subsequent track analysis and the generation of direction and velocity data were carried out as previously described (https://github.com/umkich/organelle_transport_analysis; Gavrilova et al., 2024). The supplemental movies were exported as AVIs in ImageJ, then compressed and converted into MP4 format using the H.264 encoder in HandBrake (version 1.9.2, handbrake.fr) with the contrast quality set at 22.

## Analysis of steady-state images

3D stacks with a 0.34 µm step size were generated by setting the top and bottom positions of the neurite and capturing Z-frames in between at a 50 ms exposure time. For each image, a segmented line was superimposed over the sum intensity projection in Fiji, the intensity profile plotted, and the raw grey values exported. Due to variations in the number of Z-frames captured for each image and varying levels of background autofluorescence from the gut depending on orientation of the worm, grey values were normalised by subtracting the median from each value for each image, allowing visualisation of intensity peaks over the baseline. Analysis of vesicle puncta (puncta per µm and mean peak intensity for each worm) was performed by a custom R script using the pracma package (v2.4.6) (https://cran.r-project.org/web/packages/pracma/index.html; accessed January 2026). The code for analysis of vesicle puncta is available from https://github.com/jcherzig/ida1_vesicle_analysis/tree/main.

## Statistical analysis

DCV velocity distributions were compared to control distributions by the pairwise two-sample Kolmogorov–Smirnov (K-S) test in every case with ≥50 data points. The null hypothesis that the two distributions come from the same population was rejected at $P > 0.05$. Velocities were also compared using a Kruskal–Wallis test followed by a Dunn's multiple comparisons test, as not all groups were normally distributed.

Statistical analysis for immunoblots was performed using one-way ANOVA tests followed by a post-hoc Dunnett test. For the crawling and swimming assays, normality was assessed for each sample group using a Shapiro–Wilk normality test. In all assays, some groups were not normally distributed. Therefore, statistical analysis was performed using a non-parametric Kruskal–Wallis test followed by a Dunn's multiple comparisons test. Statistical analysis for steady-state images (puncta per µm and mean peak intensity) was performed using two-tailed unpaired *t*-tests.

## Acknowledgements

We are grateful to Howard Davidson (University of Colorado Health Sciences Center, Denver), Frank McNally (University of California, Davis) and the CGC for providing worm strains. The CGC is funded by the NIH Office of Research

Infrastructure Programs (P40 OD010440). The 3i spinning disk microscope used in this study was purchased by the University of Manchester Strategic Fund. Special thanks go to Peter March for his help with imaging and to Dhanya Cheerambathur for assistance with genetic modification of worms. Finally, we are grateful to Martin Lowe for comments on the manuscript, Quentin Roebuck for technical support and InVivo Biosystems for generation of worm strains.

**Competing interests**
The authors declare no competing or financial interests.

**Author contributions**
Conceptualization: G.B.P., V.J.A.; Data curation: A.H.B., A.G.; Formal analysis: A.H.B., A.G., J.C.H.; Funding acquisition: G.B.P., V.J.A.; Investigation: A.H.B.; Methodology: A.H.B., A.G., J.C.H., G.B.P.; Project administration: G.B.P., V.J.A.; Supervision: G.B.P., V.J.A.; Visualization: A.H.B.; Writing – original draft: A.H.B., V.J.A.; Writing – review & editing: A.H.B., A.G., G.B.P., V.J.A.

**Funding**
This work was funded by a Biotechnology and Biological Sciences Research Council (BBSRC) Ph.D. CASE award (BB/T008725/1), co-funded by 3i, to A.B., a BBSRC grant to V.J.A. and G.B.P. (BB/Z517574/1), and a Wellcome Trust Ph.D. studentship to A.G. (108867/Z/15/Z). Open Access funding provided by University of Manchester. Deposited in PMC for immediate release.

**Data and resource availability**
Strains OL332 and OL370 are available from the CGC. Other new strains are available upon request. Source data not included in the supplementary material are available on FigShare (doi:10.48420/31057036). All other relevant data and details of resources can be found within the article and its supplementary information.

**First Person**
This article has an associated First Person interview with the first author of the paper.

**Peer review history**
The peer review history is available online at https://journals.biologists.com/jcs/lookup/doi/10.1242/jcs.264245.reviewer-comments.pdf

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
