## [Peer Review File · Journal of Cell Science]

Auxin-inducible degradation of UNC-116 in *C. elegans* inhibits bidirectional dense core vesicle transport and worm locomotion on different timescales

Astrid H. Boström, Anna Gavrilova, James C. Herzig, Gino B. Poulin and Victoria J. Allan
DOI: 10.1242/jcs.264245

Editor: Subhojit Roy

Review timeline

Original submission:	25 June 2025
Editorial decision:	26 August 2025
First revision received:	26 January 2026
Accepted:	12 February 2026

Original submission

First decision letter

MS ID#: jcs.264245

MS TITLE: Auxin-inducible degradation of UNC-116 in *C. elegans* inhibits bidirectional dense core vesicle transport and worm locomotion on different timescales

AUTHORS: Astrid H Boström; Anna Gavrilova; Gino B Poulin; Victoria J. Allan
ARTICLE TYPE: Research Article

Dear Dr Allan,

We have now reached a decision on the above manuscript.

Reviewer 1

Advance summary and potential significance to field

In the current manuscript the authors developed and validated an inducible kinesin-1 depletion tool for *C. elegans*. Although not completely new, the authors added an ALFA tag and performed comprehensive depletion analysis. Using this tool they observed that DVC trafficking largely stops after a few hours, whereas coordination defects need a little longer depletion. Moreover, they find differences in recovery once removed from the auxin. Although kinesin-1 being important for DVC trafficking and proper coordination is not new, now the authors are able to start depletion at adult stage showing that the phenotypes are not secondary to neurodevelopmental defects. Moreover, they show that the coordination defects are also observed upon neuron-specific depletion, showing that kinesin-1 is needed in neurons for proper functioning.

Comments for the author

Major comments

The difference in kinesin-1 depletion time needed to lead to transport defects and coordination defects is interesting. Although the potential reasons for this are not addressed experimentally, the

authors discuss this in the discussion. Yet these findings raise many questions. How does kinesin-1 depletion affect the DCV distribution? How general are these findings? e.g. the authors only looked at DCV transport in the ALA neuron. Are other unc-116 cargoes similarly affected and do other neurons show the same phenotype. And how about other motors? The motors seem to show strong interdependence in functioning and localization, so depleting one motor may have a strong defect on the other. It would be very interesting to look at some non-unc-116 cargo and/or dynein functioning + localization. While I can understand that some of these points may be outside the scope of this manuscript, I feel that some extra experiments are needed to validate the generality of the findings. Easiest being to look whether other neurons show the same DCV dynamics defect upon kinesin-1 depletion and how the overall (steady state) DCV distribution is affected by kinesin-1 depletion.

Moreover the results summary states "although DCV movement recovered well after removal of auxin, recovery of locomotion was limited after long-term UNC-116 degradation over the time course analyzed." If I'm not mistaken, the auxin treatment times were not equal and you have shown in this manuscript that the length of auxin exposure is important. If this is indeed the case than making this comparison is not fair. Experiments should be performed in the same manner to be able to compare recoveries.

Minor comments

- line 49: Add paper shen lab: Yan et al 2013
- Line 330 "imply that kinesin-1/UNC-116 is the main anterograde motor for DCV transport in *C. elegans*, contrary to previous reports that kinesin-3/UNC-104 plays a critical role" I fail to see how a role for unc-104 was excluded. Previous reports have shown that also unc-104 plays an important role.
- For many *C. elegans* figures it is not clear how many animals are analyzed and at what age
- Figure 3I-J auxin-labeling not clear
- Figure 2+6 experimental design is not explained in figure. And Auxin / K-NAA is used interchangeable (Figure 5 and 7).

Reviewer 2

Advance summary and potential significance to field

Authors have commendable study that used temporal knockdown using the auxin-degron system to uncover specific roles of the Kinesin-I in vivo. There are several studies on Kinesin-I roles in axonal transport. Using general mutations have always been problematic due several developmental effects and specifically for Kinesin and Dynein their roles in microtubule orientation. This elegant study thus brings insight into a thorny problem-what are the specific roles of multiple motors for the transport of single defined cargo. The study would benefit if some concerns can be addressed.

Comments for the author

Major comments [Please request additional experiments only if they are essential for supporting the conclusions; authors should be encouraged to highlight any claims that are preliminary or speculative, or to discuss any pitfalls or alternative interpretations in a 'Limitations' section]

(i) There is no real link is provided between the dense core vesicle transport analysis and the behaviour experiments. I think authors should consider moving this earlier in the paper before the description of axonal transport phenotypes. If the authors are suggesting that IDA-1 transport and potentially DCV transport is important for worm locomotion-this is not well substantiated. It is well established the synaptic vesicle transport is essential for locomotion. If co-relations need to be drawn it is important to assess synaptic vesicle transport.

(ii) To show DCV transport is affected it would be useful to one additional DCV marker-perhaps just steady state distributions to ensure that the IDA-1 marker captures say a neuropeptide cargo.

(iii) Please add the neuronal auxin treatment data in the main figures. I think there needs to be discussion on the differences between ubiquitous knockdown and neuronal knockdown. Do you think some of the effects are non-cell autonomous? This possibility needs to be discussed and potentially experimentally addressed. Does the neuronal knockdown alone also show a reduction in UNC-116 levels?

(iv) It would be good to present more data which includes runlength, pauses and potentially reversals. Velocity is one of the hardest parameters to understand wrt motor numbers and mechanisms.

(v) It might be good to include orientation of microtubules in the neuronal and ubiquitous knockdown of the motor. This might be useful to exclude. Is there any data on if the levels of other motors namely dyenin and unc-104/KIF1A are unchanged in the Kinesin-I knockdown? I think both the reversals, dyenin mutants will speak to the co-ordination with Kinesin-I in *ida-1* transport especially for the co-ordination between motors interpretations that are mentioned throughout the manuscript.

Minor comments

(i) Please list the number of animals imaged and number of vesicles imaged.

First revision

Author response to reviewers' comments

We would like to thank both reviewers for their insightful and helpful comments. We believe that addressing them has significantly improved the paper. Our response to each comment is provided below. We should point out that, based on the comments of reviewer 2, we have changed the order of the results section to cover locomotion first, followed by analysis of DCV motility. This means the order of figures has been altered substantially. In the rebuttal below we use the new figure numbers rather than the old ones. The changed figure numbers are: fig. 2 is now 7, with new panel D; fig. 5 is now 2; fig. 6 is now fig. S6; fig. 7 is now 3. Any additional figures or figure panels are referred to as 'new figure x', for clarity. In the pdf where text changes are summarised, we have chosen not to show moved text, as that would be very hard to follow. Instead, we highlight any added text. We also don't show any text that was removed during editing for manuscript length and clarity, unless it was to respond to a specific reviewer's comment.

Reviewer 1:

Major comments

The difference in kinesin-1 depletion time needed to lead to transport defects and coordination defects is interesting. Although the potential reasons for this are not addressed experimentally, the authors discuss this in the discussion. Yet these findings raise many questions. How does kinesin-1 depletion affect the DCV distribution? How general are these findings? e.g. the authors only looked at DCV transport in the ALA neuron. Are other unc-116 cargoes similarly affected and do other neurons show the same phenotype. And how about other motors? The motors seem to show strong interdependence in functioning and localization, so depleting one motor may have a strong defect on the other. It would be very interesting to look at some non-unc-116 cargo and/or dynein functioning + localization. While I can understand that some of these points may be outside the scope of this manuscript, I feel that some extra experiments are needed to validate the generality of the findings. Easiest being to look whether other neurons show the same DCV dynamics defect upon kinesin-1 depletion and how the overall (steady state) DCV distribution is affected by kinesin-1 depletion.

We thank the reviewer for their enthusiasm and suggestions. While many of these fascinating

questions are to be investigated in the future (e.g. the relationship with dynein, how other kinesin-1 cargoes are affected in other neurons, and what happens to DCV motility when UNC-104 is depleted), we have addressed a number of these points.

1. The important question of whether DCVs labelled with other markers in different neurons behave similarly to the IDA-1::GFP-labelled DCVs in the ALA neuron has been addressed using a strain from Sieburth et al. (*Nature* 436, 510-517, 2005) expressing a DCV cargo, NLP-21::VENUS, in a subset of DB motor neurons. We imaged their motility in the DB7 neuron and find that their bi-directional movement in the DB7 commissure is strongly inhibited (new fig. 6B, new movie 6). Their steady-state distribution in the DB7 axon is slightly shifted towards the cell body, but the most obvious change is that their intensity greatly increases (new fig. 6C and D). One possible explanation for this is that their fusion with the cell membrane is reduced without UNC-116. Why this happens is something we will investigate in the future. In addition, we used the same neuron to assess the steady state distribution of the pre-synaptic/synaptic vesicle marker GFP::SNB-1 (strain from Sieburth et al.) and found no changes when UNC-116 was degraded (new fig. 6E, F). Together, these new data show the effect of UNC-116 loss on DCVs is seen in two different neuron types with two different types of marker, while the distribution of synapses is not affected. The relevant new text is in lines 258-276.
2. We also looked at the effects on steady state distribution of IDA-1::GFP-labelled DCVs in the ALA neuron after 24 hours in K-NAA, and this provided important new information (new fig. 5; new supplementary figures 4 and 5). While there was no change in DCV distribution or intensity in the proximal ALA axon (fig. 5B, C, fig. S4), the distribution of DCVs in the distal axon changed significantly (fig. D,E, fig. S5). The number of DCVs, but not their intensity, changed in the bulk of the distal axon, suggesting a depletion of DCVs from that region. Strikingly, there was a build-up of DCV fluorescence right at the axon tip. Lines 246-257 in the results describe these results. Taken together, this new data suggests that there may be some continued dynein-driven motility (probably early during depletion) that removes some DCVs, and that UNC-104 may carry some DCVs right to the axon tip. We have discussed these possibilities in lines 433-470.

Moreover the results summary states "although DCV movement recovered well after removal of auxin, recovery of locomotion was limited after long-term UNC-116 degradation over the time course analyzed." If I'm not mistaken, the auxin treatment times were not equal and you have shown in this manuscript that the length of auxin exposure is important. If this is indeed the case than making this comparison is not fair. Experiments should be performed in the same manner to be able to compare recoveries.

Thank you for pointing out this major flaw in our argument. We have addressed this by repeating the experiment with a 10-hour K-NAA treatment followed by rescue on plates without K-NAA, then analysing UNC-116 levels (new panel in fig. 7D), swimming, crawling and DCV motility (new fig. 8). This is covered in new results text (lines 277-331) and the discussion (lines 405-408). This experiment gave very interesting results. While DCV motility recovered somewhat, it was not back to the near normal levels we saw after rescue from a 4-hour K-NAA treatment (fig. 4). In contrast, crawling was reduced following 10-hour K-NAA treatment and recovered completely after 24 hours without K-NAA (fig. 8C). Swimming, on the other hand, was not significantly affected after the initial treatment, but then declined sharply over the next 24 or 48 hours without K-NAA. The reason for this continued decline remains to be identified, but the data show that swimming and crawling respond very differently to UNC-116 degradation, backing up our original observations that the time of onset of inhibition of swimming and crawling differed (fig. 2).

Minor comments

- line 49: Add paper shen lab: Yan et al 2013.

Done. We have also added extra references to the text.

- Line 330 "imply that kinesin-1/UNC-116 is the main anterograde motor for DCV transport in C. elegans, contrary to previous reports that kinesin-3/UNC-104 plays a critical role" I fail to see how a role for unc-104 was excluded. Previous reports have shown that also unc-104 plays an important role.

We have rewritten this text in line 468-470: “Altogether, our data indicate that although kinesin-3/UNC-104 is active throughout the length of the axon in *C. elegans*, UNC-116 is the dominant anterograde motor once DCVs leave the cell body.”. We have also revised our discussion of the role of UNC-104 based on our new data (discussion lines 457-479)

- For many C. elegans figures it is not clear how many animals are analyzed and at what age

This information was in the supplementary tables, but as this was rather obscure, we have now added the number of animals and their age in each figure or figure legend, as appropriate. The number of kymographs and motile segments analysed is only listed in the supplementary tables, which are referred to in the figure legends.

- Figure 3I-J auxin-labeling not clear

We have clarified this (fig. 3 is now fig. S1)

- Figure 2+6 experimental design is not explained in figure. And Auxin / K-NAA is used interchangeable (Figure 5 and 7).

We have made some improvements to presentation and included experimental design summaries throughout, which we hope deals with this issue. We have also made sure to use K-NAA instead of auxin throughout the results text and figures, and most of the discussion.

Reviewer 2:

Major comments

(i) There is no real link is provided between the dense core vesicle transport analysis and the behavior experiments. I think authors should consider moving this earlier in the paper before the description of axonal transport phenotypes. If the authors are suggesting that IDA-1 transport and potentially DCV transport is important for worm locomotion-this is not well substantiated. It is well established the synaptic vesicle transport is essential for locomotion. If co-relations need to be drawn it is important to assess synaptic vesicle transport.

We agree that DCVs are not the only cargo that kinesin-1 may be transporting that could influence locomotion. We have expanded upon this in the discussion (lines 352-397). We have followed the reviewer’s suggestion to switch the order of the results, and now cover the changes in locomotion first, followed by DCV motility (as an exemplar UNC-116 cargo). However, it is important to stress that DCVs are known to be important for locomotion (not just synaptic vesicles) and so we have provided more information in the discussion (lines 380-397). We have now imaged GFP::*SNB-1*, a synaptic vesicle marker, and find that its localisation in the axon of the DB7 motoneuron isn’t affected by UNC-116 degradation (new fig. 6E, F).

(ii) To show DCV transport is affected it would be useful to one additional DCV marker- perhaps just steady state distributions to ensure that the IDA-1 marker captures say a neuropeptide cargo.

Thank you for this suggestion, and as described above, we find that motility of DCVs marked by a neuropeptide cargo are similarly inhibited by loss of UNC-116 (new fig. 6B, new movie 6). Strikingly, the DCV puncta in the axon become much brighter after UNC-116 degradation, suggesting that their secretion is blocked. We will investigate the mechanism for this further in the future.

(iii) Please add the neuronal auxin treatment data in the main figures. I think there needs to be discussion on the differences between ubiquitous knockdown and neuronal knockdown. Do you think some of the effects are noncell autonomous? This possibility needs to be discussed and potentially experimentally addressed. Does the neuronal knockdown alone also show a reduction in UNC-116 levels?

The neuronal degradation data was already in the main figures (fig. 4), with the somatic degradation strain in fig. S2 (new fig. S3). We have improved the figure legends to make this clearer and checked the results text also. We hope this is sufficient. Since the neuronal and ubiquitous degradation of UNC-116 gives very similar phenotypes for DCV motility, swimming and crawling, we think there are only minor (if any) non-cell autonomous effects. This is mentioned in the discussion (lines 483-487). The neuronal degradation does give a small decrease in UNC-116 levels, as was shown in fig. 1C, but it was not statistically significant. This change was as expected, given that UNC-116 is highly expressed in neurons (therefore a decrease was seen), but there is a bigger pool of the motor in other cells, which remains intact.

(iv) It would be good to present more data which includes run length, pauses and potentially reversals. Velocity is one of the hardest parameters to understand wrt motor numbers and mechanisms.

We agree about velocity being hard to interpret. However, it would not be valid to assess run length as many DCV runs start and/or end outside the field of view, therefore would not give a true run length. This is now mentioned in the results text (lines 211-213). Pauses are quite rare, as are reversals, and given that the number of movements drops precipitously upon UNC-116 degradation, there would be very few examples. In addition, we have looked again at all the kymographs and don't see any new behaviour after degradation for 1 hour.

(v) It might be good to include orientation of microtubules in the neuronal and ubiquitous knock down of the motor. This might be useful to exclude. Is there any data on if the levels of other motors namely dynein and unc104/KIF1A are unchanged in the Kinesin-I knockdown? I think both the reversals, dynein mutants will speak to the co-ordination with Kinesin-I in ida-1 transport especially for the co-ordination between motors interpretations that are mentioned throughout the manuscript.

We have tried to determine the MT orientation in the ALA neuron by generating an EBP-2 strain but see no MT comets in the ALA, although they're clearly visible in other neurons. That has been added in the results text (lines 199-204). In any case, both directions of DCV motility are affected profoundly, and the small number of remaining outward movements and the accumulation of a small number of DCVs right at the axon tip (new fig. 5, new figs S4, S5) would fit with that being driven by UNC-104. Unfortunately, due to lack of antibodies that work in worms the levels of UNC-104 and dynein can't be assessed biochemically. We have assessed a synaptic cargo, GFP::SNB-1, and find that its distribution is not altered by UNC-116 degradation (new fig. 6E, F), suggesting that UNC-104 activity is not affected.

Minor comments

(i) Please list the number of animals imaged and number of vesicles imaged.

Please see our response to reviewer 1. The animal numbers are included in the figures and the number of vesicles/tracks analysed is provided in the supplementary tables. Note that we used the software KymoButler to analyse kymographs. KymoButler outputs tracks that usually correspond to the movement of one vesicle. However, the algorithm is not entirely perfect and a single DCV may sometimes give rise to more than one track (in part due to the large number of stationary tracks interrupting moving tracks). Thus, we display "number of tracks" rather than "number of vesicles".

Second decision letter

MS ID#: jcs.264245R1

MS Title: Auxin-inducible degradation of UNC-116 in *C. elegans* inhibits bidirectional dense core vesicle transport and worm locomotion on different timescales

Authors: Astrid H Boström; Anna Gavrilova; James C. Herzig; Gino B Poulin; Victoria J. Allan

Article Type: Research Article

Dear Dr Allan,

I am happy to tell you that your manuscript has been accepted for publication in Journal of Cell Science, pending standard publication integrity checks.